# Initial phospholipid-dependent Irgb6 targeting to *Toxoplasma gondii* vacuoles mediates host defense

Youngae Lee[1,3], Hiroshi Yamada[5], Ariel Pradipta[1], Ji Su Ma[1,3], Masaaki Okamoto[1], Hikaru Nagaoka[6], Eizo Takashima[6], Daron M Standley[2,4], Miwa Sasai[1,3], Kohji Takei[5], Masahiro Yamamoto[1,3]

***Toxoplasma gondii* is an obligate intracellular protozoan parasite capable of infecting warm-blooded animals by ingestion. The organism enters host cells and resides in the cytoplasm in a membrane-bound parasitophorous vacuole (PV). Inducing an interferon response enables IFN-γ–inducible immunity-related GTPase (IRG protein) to accumulate on the PV and to restrict parasite growth. However, little is known about the mechanisms by which IRG proteins recognize and destroy *T. gondii* PV. We characterized the role of IRG protein Irgb6 in the cell-autonomous response against *T. gondii*, which involves vacuole ubiquitination and breakdown. We show that Irgb6 is capable of binding a specific phospholipid on the PV membrane. Furthermore, the absence of Irgb6 causes reduced targeting of other effector IRG proteins to the PV. This suggests that Irgb6 has a role as a pioneer in the process by which multiple IRG proteins access the PV. Irgb6-deficient mice are highly susceptible to infection by a strain of *T. gondii* avirulent in wild-type mice.**

## Introduction

Healthy mammalian hosts activate immune responses against pathogenic infections. The innate immune response first induces IL-12 production by antigen-presenting cells, such as macrophages and dendritic cells. This is carried out via recognition of pathogen-derived components by microbe pattern recognition receptors such as toll-like receptors (Hunter & Remington, 1995; Yarovinsky & Sher, 2006). IL-12 subsequently stimulates the antipathogen type I immune response, wherein naïve CD4[+] or CD8[+] T cells become antigen-specific Th1 cells and cytotoxic T cells, respectively, with the help of antigen-presenting cells. Th1 cells, cytotoxic T lymphocytes, and natural killer cells produce IFN-γ to activate the various cell-autonomous programs targeting vacuolar pathogens (Suzuki et al, 1988; Gazzinelli et al, 1991).

One of the IFN-γ–induced cell-autonomous programs is associated with IFN-inducible GTPases, such as p47 immunity-related GTPases (IRGs) and p65 guanylate-binding proteins (GBPs) (Kim et al, 2012). Immunity-related GTPases and GBPs belong to the dynamin GTPase superfamily (Martens & Howard, 2006; Pawlowski, 2010; Kim et al, 2012) and can target wide ranges of bacterial, fungal, and protozoan vacuolar pathogens (Coers et al, 2008; Al-Zeer et al, 2009; Ferreira-da-Silva Mda et al, 2014; Kuriakose & Kanneganti, 2017; Santos & Broz, 2018). In mice, the IRG protein family consists of three regulator IRG proteins (Irgm1, Irgm2, and Irgm3) and over 20 effector IRG proteins and decoys (Bekpen et al, 2005; Muller & Howard, 2016). There are four effector IRG proteins known to be expressed in mice: Irga6, Irgb6, Irgb10, and Irgd (Martens & Howard, 2006). Regulator IRG proteins harboring $GX_4GMS$ in the first nucleotide-binding motif (G1) are mainly associated with host endomembranes, such as the Golgi apparatus and ER (Bekpen et al, 2005; Hunn et al, 2011). Effector IRG proteins possess a universally conserved $GX_4GKS$ sequence in the G1 motif, enabling binding to both GTP and GDP (Taylor et al, 1996; Uthaiah et al, 2003; Bekpen et al, 2005; Hunn et al, 2008). The GTPase activity has been demonstrated for Irga6 and Irgm3 (Taylor et al, 1996; Uthaiah et al, 2003; Hunn et al, 2008). Regulator IRG proteins can maintain effector IRG proteins in an inactive GDP-bound state, potentially preventing the latter from inappropriate activation on host cell membrane–bounded vesicular systems. In their absence, effector IRG proteins likely form GTP-bound aggregates and are unable to interact with the *Toxoplasma gondii* parasitophorous vacuole (PV) (Martens et al, 2004; Hunn et al, 2008; Hunn & Howard, 2010; Coers, 2013; Haldar et al, 2013). There are 11 members in the mouse GBP family, all of which have the conserved GTP binding motifs (Kresse et al, 2008). Guanylate-binding protein mutants lacking GTPase activity are incapable of accumulating at *T. gondii* PV membrane (PVM) (Degrandi et al, 2013; Ohshima et al, 2015). When these IFN-inducible GTPases are recruited to the PVM, it becomes vesiculated and disrupted, resulting in death of the vacuolar pathogen (Martens et al, 2005; Ling et al, 2006; Degrandi et al, 2007; Virreira Winter et al, 2011; Yamamoto

[1]Department of Immunoparasitology, Research Institute for Microbial Diseases, Osaka University, Osaka, Japan   [2]Department of Genome Informatics, Research Institute for Microbial Diseases, Osaka University, Osaka, Japan   [3]Laboratory of Immunoparasitology, WPI Immunology Frontier Research Center, Osaka University, Osaka, Japan   [4]Laboratory of Systems Immunology, WPI Immunology Frontier Research Center, Osaka University, Osaka, Japan   [5]Department of Neuroscience, Graduate School of Medicine, Dentistry and Pharmaceutical Sciences, Okayama University, Okayama, Japan   [6]Division of Malaria Research, Proteo-Science Center, Ehime University, Ehime, Japan

Correspondence: myamamoto@biken.osaka-u.ac.jp

et al, 2012; Selleck et al, 2013). Thus, GTPase activity–dependent IRG and GBP accumulation is well established as important for cell-autonomous immunity to vacuolar pathogens.

The mechanism by which IRG proteins access *T. gondii* PV from the cytosolic compartments can be passive. This process depends on diffusion from the cytoplasmic pools rather than active transport involving toll-like receptor–mediated signaling pathways or microtubule networks (Khaminets et al, 2010). Although IRG proteins are localized on the PVM within a few minutes of *T. gondii* infection (Hunn et al, 2008; Khaminets et al, 2010), little is known about the mechanism by which IRG proteins recognize and destroy the PVM thus far. This process is important for IFN-γ–induced cell-autonomous immunity. Among the effector IRG proteins, Irgb6 and Irgb10 are loaded first and most efficiently onto *T. gondii* PVM (Khaminets et al, 2010).

Here, we aimed to determine the role of Irgb6 in the cell-autonomous response against *T. gondii*, mediating ubiquitination and disruption of *T. gondii* PVM.

# Results

### Irgb6 contributes to IFN-γ–induced cell-autonomous resistance to *T. gondii*

Several studies using gene-deficient mice (Taylor et al, 2000, 2007; Liesenfeld et al, 2011) have shown that Irgm1 (also called LRG-47), Irgm3 (IGTP), Irga6 (IIGP, IIGP1), and Irgd (IRG-47) have critical roles in the anti–*T. gondii* response. Earlier studies have shown that Irgb6 and Irgb10 proteins function as pioneers, loading first to the *T. gondii* vacuoles (Khaminets et al, 2010). However, there are no reports clearly showing that Irgb6 has a protective role for the host against intracellular microbial pathogens using mice in vivo lacking endogenous Irgb6. To determine the role of Irgb6 in host protective immunity against *T. gondii* infection, we generated Irgb6-deficient mice via CRISPR/Cas9 genome editing (Fig 1A). The Irgb6 locus in C57BL/6 mice contains two Irgb6 genes (Irgb6* and Irgb6), both of which encode identical amino acid sequences despite their nucleotide coding sequences differing at four positions (Bekpen et al, 2005). To target both Irgb6 genes, we designed a shared gRNA (Table S1) for both Irgb6* and Irgb6 (Fig 1A). We obtained Irgb6-deficient MEFs from fetuses lacking both Irgb6* and Irgb6 proteins. The level of protein expression of other IFN-γ–inducible GTPases was measured by Western blotting in Irgb6-deficient MEFs and was comparable with those in wild-type MEFs (Fig 1B). Sequence analysis of both Irgb6* and Irgb6 genomic DNA from Irgb6-deficient MEFs revealed an 891-bp deletion (position 172–1062) in the coding region of the genes (Fig S1A and B). To test whether the Irgb5-b4 tandem (decoy) gene lies between the two mutated Irgb6 genes, the coding regions of IFN-γ–treated Irgb6-deficient MEF cDNA were amplified by PCR (Fig S2A). Direct sequencing of the 3′ region of the coding sequence showed that the PCR product contained the coding sequence derived from Irgb3 and Irgb4 genes. These genes could not be individually amplified because of their high sequence similarity. The DNA sequences at positions 2208T, 2286A, and 2307T indicate the presence of the Irgb4 gene (Bekpen et al, 2005) (Fig S2B). The PCR products were then cloned into cloning vectors and

individually sequenced. The results indicated that the coding region (1–2,535 bp) of the Irgb5-b4 gene is normally present between the two mutated Irgb6 genes (Fig S2C). In addition, the 3′ untranslated region of Irgb4 was normally detected in Irgb6 knockout MEFs (Fig S2D), indicating that Irgb5-b4 is still present and unaffected by genome editing. In a *T. gondii* killing assay, our data showed that Irgb6 deficiency causes significantly reduced *T. gondii* killing activity compared with wild-type and Irgb10 single-deficient MEFs. Irgb6/Irgb10 double-deficient MEFs displayed slightly reduced *T. gondii* killing compared with Irgb6 single-deficient cells (Fig 1C), suggesting that Irgb6 has a more dominant role in *T. gondii* killing activity in vitro compared with Irgb10. However, compared with Irgm1/m3 double-deficient cells (IRG effector activity may be more completely lost), Irgb6-deficient cells had partially reduced *T. gondii* killing activity (Fig 1D). This may be caused by other residual IRG effector proteins, as well as GBP proteins, on the PVM in Irgb6-deficient cells (Fig 2A–E). To test whether Irgb6 is involved in IFN-γ–mediated PVM disruption, we performed electron microscopy to analyze IFN-γ–stimulated wild-type and Irgb6-deficient MEFs infected with *T. gondii*. The data displayed that *T. gondii* vacuole vesiculation and disruption were clearly observed in IFN-γ–treated wild-type MEFs (Martens et al, 2005) (Fig 1E) compared with those in Irgb6-deficient cells (Fig 1F). This indicated that Irgb6 is important for IFN-γ–induced PVM disruption.

We also found that compared with wild-type BMDMs, BMDMs from Irgb6-deficient mice displayed impaired *T. gondii* killing activity (Fig S3B), increased infection rate (Fig S3C), and reduced recruitment of Irgb10, Irga6, Gbp1, Gbp2, Gbp1-5, ubiquitin, and p62 (Fig S3D). Altogether, these results show that Irgb6 is required for the IFN-γ–induced anti–*T. gondii* cell-autonomous response in vitro.

### Irgb6 is required for recruitment of ubiquitin and other IFN-inducible GTPases to *T. gondii* PVM

We next examined whether Irgb6 is required for normal loading of other effector IRG proteins and GBPs onto the *T. gondii* PVM using immunofluorescence staining (Fig 2A–E). Compared with wild-type cells, Irgb6-deficient MEFs displayed significantly reduced accumulation of Irga6, Gbp1, Gbp2, Gbp1-5, and Irgb10 on the *T. gondii* PVM (Fig 2A–E). In Irgb6-deficient MEFs, this cannot be due to reduced protein expression (Fig 1B). Interestingly, p62 and ubiquitin loading onto the PVM in Irgb6-deficient MEFs was also severely impaired (Fig 2F and G). Moreover, almost all ubiquitin colocalized with Irgb6 on the *T. gondii* PVM. A low ratio of Irgb6 single-positive vacuoles was detectable, whereas ubiquitin single-positive vacuoles were almost undetectable. This suggests that ubiquitination of the *T. gondii* PVM is completely dependent on Irgb6 (Fig 2H and I). Together, our findings suggest that Irgb6 accumulation on the *T. gondii* PVM mediates loading of ubiquitin and other IFN-γ–inducible GTPase effectors on the PVM. This leads to enhanced PVM disruption, followed by early killing of the parasite through collaborative interactions with other effectors.

### Other effector IRG proteins are not involved in Irgb6 and ubiquitin loading onto *T. gondii* vacuoles

In response to *T. gondii* infection, Irgb6 and Irgb10 have been shown to be initially localized on the PVM within a few minutes of infection

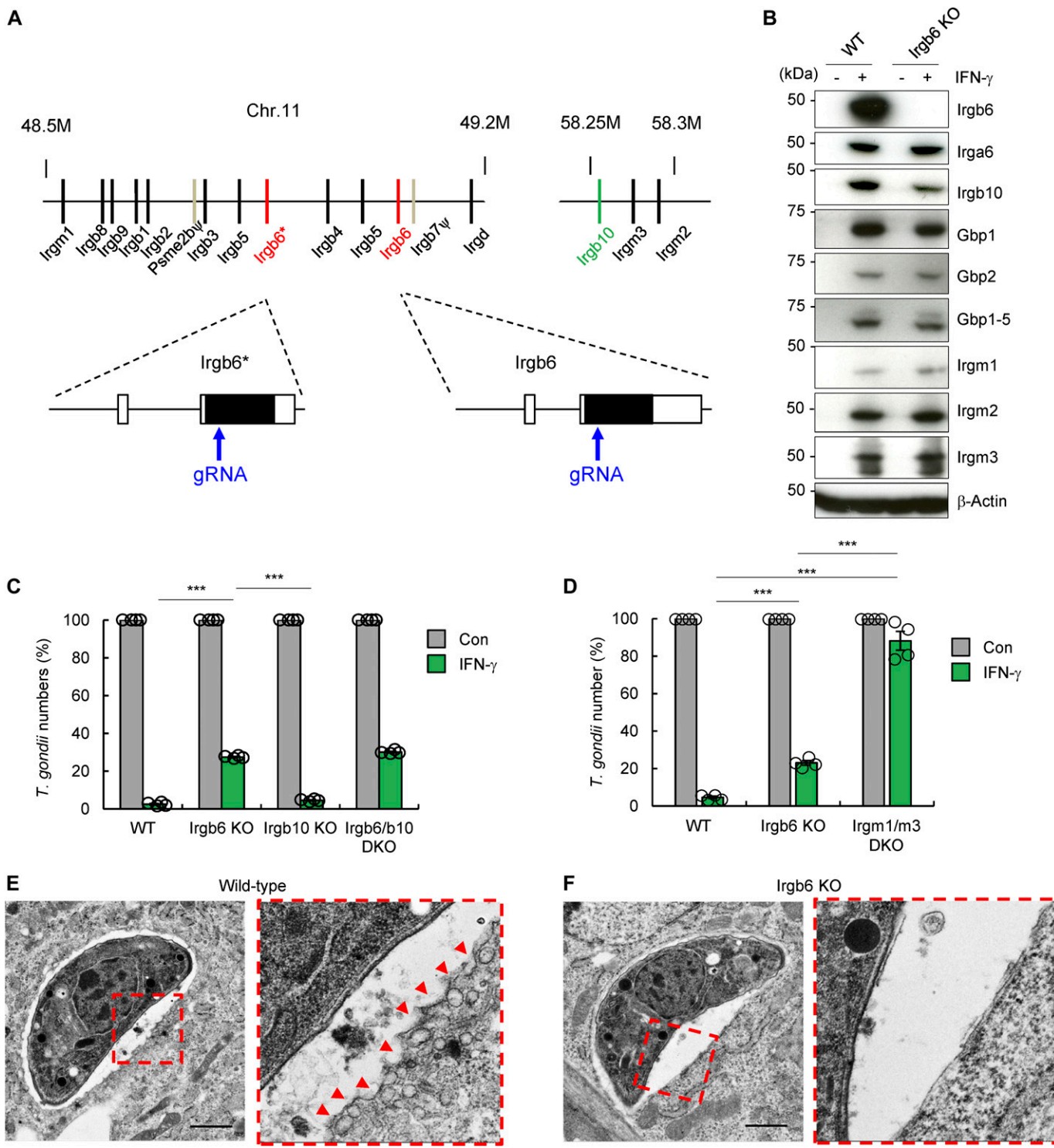

**Figure 1. Irgb6 significantly contributes to IFN-γ–induced cell-autonomous *T. gondii* killing.**
**(A)** Schematic representation of the gene-targeting strategy for mouse Irgb6* and Irgb6 locus by Cas9-mediated genome editing. **(B)** Western blot analysis of the indicated protein expressions in WT and Irgb6 KO MEFs after IFN-γ stimulation or not. **(C, D)** Survival rate of *T. gondii* in the presence of IFN-γ stimulation relative to that in the non–IFN-γ–treated control by luciferase analysis at 24 h postinfection. The graphs show the mean ± SEM in four independent experiments. Two-tailed *t* tests were used: ***P < 0.001 versus WT, Irgb10 KO MEFs or Irgm1/m3 DKO MEFs. **(E, F)** Electron microscope images of *T. gondii*–infected WT (E) and Irgb6 KO (F) MEFs at 4 h postinfection in the presence of IFN-γ stimulation. The images are representative of three independent experiments. Red arrowheads indicate membrane blebbing. Scale bars, 1 μm.

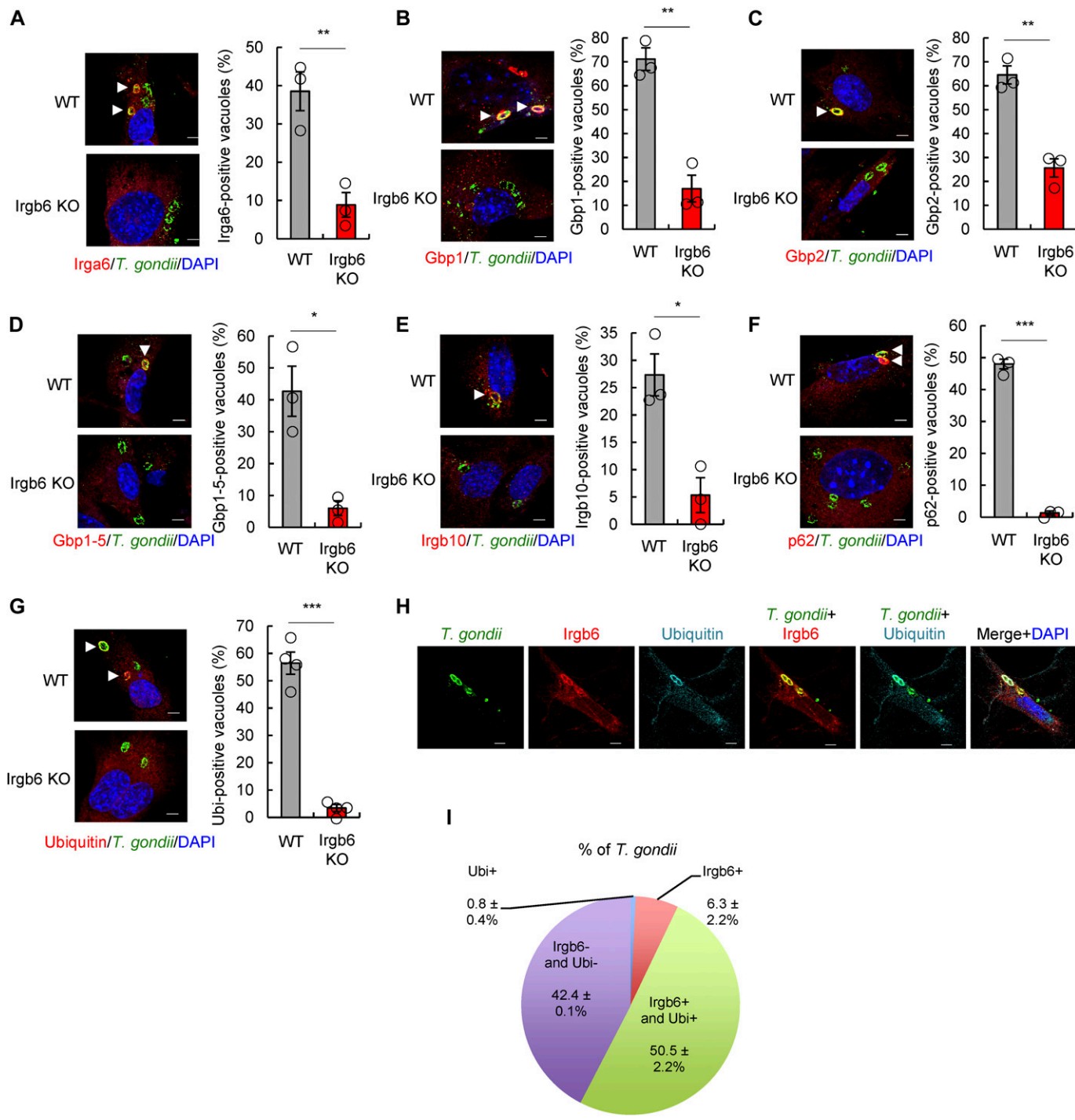

**Figure 2. Irgb6 recruits other IFN-γ–inducible GTPases and ubiquitin on the *T. gondii* PVM.**
**(A, B, C, D, E, F, G)** Confocal microscope images (left) and the graphs (right) represent the localization of Irga6 (A), Gbp1 (B), Gbp2 (C), Gbp1-5 (D), Irgb10 (E), p62 (F), and ubiquitin (G) (red) to *T. gondii* vacuoles (green), and DAPI (blue) at 4 h postinfection in IFN-γ–treated WT and Irgb6 KO MEFs. **(H)** Confocal microscope images represent the colocalization of Irgb6 (red) and ubiquitin (cyan blue) to *T. gondii* vacuoles (green), and DAPI (blue) at 4 h postinfection in IFN-γ–treated WT MEFs. **(I)** The pie chart represents the mean ± SEM of Irgb6 and ubiquitin double- or Irgb6 single- or ubiquitin single-positive vacuoles in IFN-γ–treated WT MEFs. All graphs show the mean ± SEM in three or four independent experiments. All images are representative of three or four independent experiments. White arrowheads indicate colocalization. Scale bars, 5 μm. Two-tailed *t* tests were used: *$P < 0.05$, **$P < 0.01$, and ***$P < 0.001$.

(Khaminets et al, 2010). As shown in Fig 2, Irgb6 is required for normal levels of ubiquitin and other effector IRG protein binding to the *T. gondii* PVM. However, Irga6 deficiency (Fig 3A and B) and

Irgb10 deficiency (Fig 3C and D) did not affect Irgb6 and ubiquitin localization on the PVM compared with wild-type MEFs. This suggests that Irgb6 and ubiquitin initially have the ability to bind to

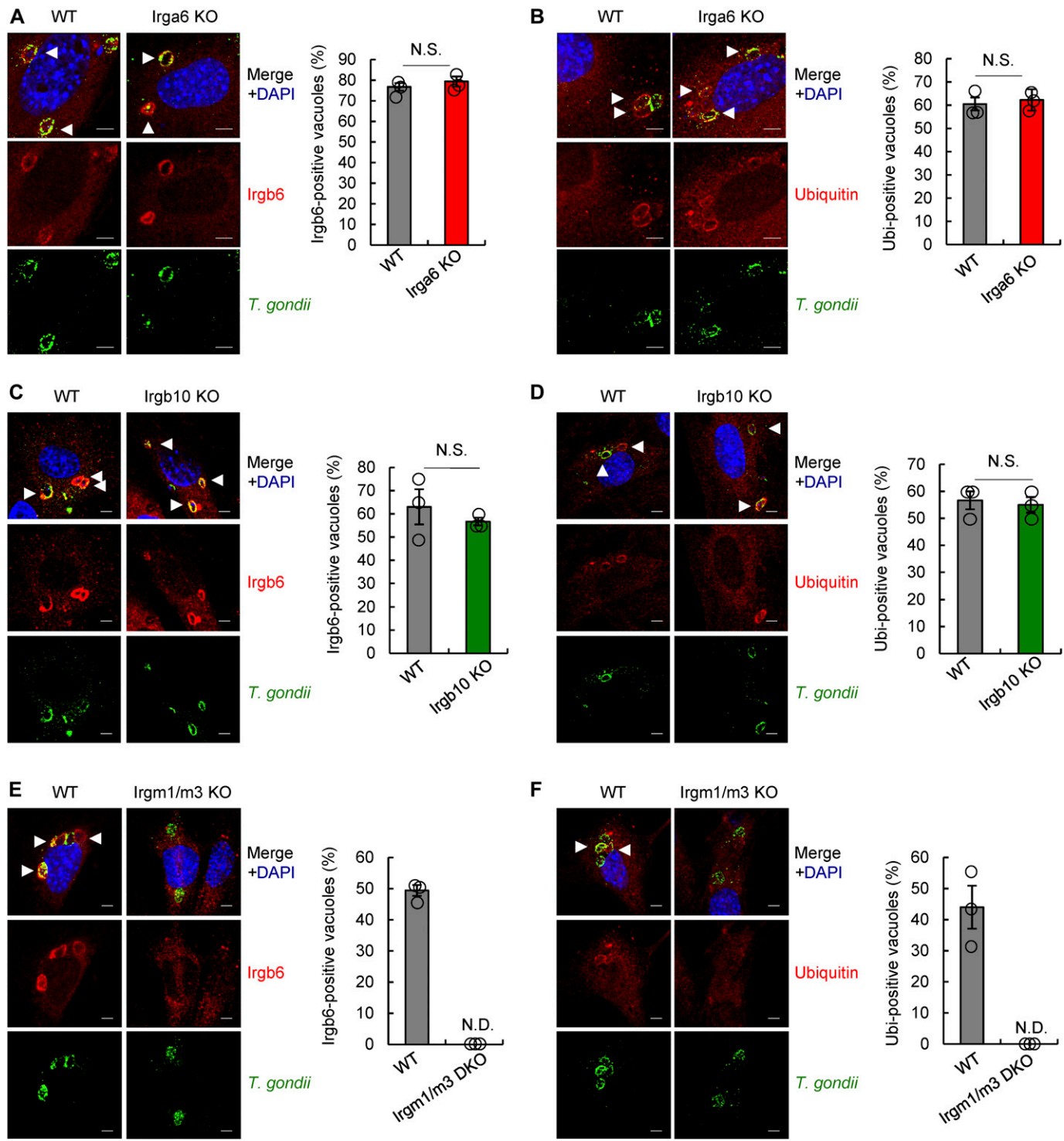

**Figure 3. Regulator IRG proteins, but not other effector IRG proteins, are required for loading of Irgb6 and ubiquitin on *T. gondii* PVM.**
**(A, B)** Confocal microscope images (left) and the graphs (right) represent the localization of Irgb6 (A) and ubiquitin (B) (red) to *T. gondii* vacuoles (green), and DAPI (blue) at 4 h postinfection in IFN-γ–treated WT and Irga6 KO MEFs. **(C, D)** Confocal microscope images (left) and the graphs (right) represent the localization of Irgb6 (C) and ubiquitin (D) (red) to *T. gondii* vacuoles (green), and DAPI (blue) at 4 h postinfection in IFN-γ–treated WT and Irgb10 KO MEFs. **(E, F)** Confocal microscope images (left) and the graphs (right) represent the localization of Irgb6 (E) and ubiquitin (F) (red) to *T. gondii* vacuoles (green), and DAPI (blue) at 4 h postinfection in IFN-γ–treated WT and Irgm1/m3 DKO MEFs. All graphs show the mean ± SEM in three independent experiments. All images are representative of three independent experiments. White arrowheads indicate colocalization. Scale bars, 5 μm. ND, not detected; NS, not significant.

*T. gondii* PVM independent of Irga6 and Irgb10. Furthermore, we investigated whether Irgb10 was capable of compensating for Irgb6 deficiency. Immunofluorescence staining identified no significant difference in Irga6, Gbp1, Gbp2, Gbp1-5, ubiquitin, and p62 recruitment on *T. gondii* PVM when observing Irgb6 single- and Irgb6/Irgb10 double-deficient MEFs (Fig S3A). This implies that only Irgb6 is required for the recruitment of other effector IRG proteins, GBPs, and ubiquitin.

### Regulator IRG proteins are required for Irgb6 and ubiquitin recruitment on *T. gondii* vacuoles

During *T. gondii* infection, regulator IRG proteins such as Irgm1 and Irgm3 have been shown to be involved in proper localization of active form, GTP-bound effector IRG proteins on the PVM. These proteins distinguish the membranes of cellular organelles from "non-self" vacuoles (Coers et al, 2008; Hunn et al, 2008; Papic et al, 2008; Haldar et al, 2013). Recruitment of Irgb10 and Gbp2 on the *T. gondii* PVM is completely unsuccessful in Irgm1/m3 double-deficient MEFs (Haldar et al, 2013). Our previous findings also showed that IFN-γ–induced ubiquitin and p62 localization on the *T. gondii* PVM is not detectable in Irgm1/m3 double-deficient MEFs (Lee et al, 2015). Thus, we examined the contribution of regulator IRG proteins to controlling Irgb6 localization on the *T. gondii* PVM using Irgm1/m3 double-deficient MEFs. A previous study has reported that regulator IRG proteins are required for the localization of Irgb6 on the PVM in a different system (Hunn et al, 2008). As expected, IFN-γ–induced localization of Irgb6 and ubiquitin on *T. gondii* PVM was completely unsuccessful in Irgm1/m3 double-deficient MEFs (Fig 3E and F).

### GTPase activity is required for Irgb6 localization at *T. gondii* PVM

We next analyzed the molecular mechanisms underlying how Irgb6 is recruited to *T. gondii* PVM. The nucleotide-bound status of Irgb6 was previously shown to be important for Irgb6 loading onto *T. gondii* PVM (Hunn et al, 2008). Irgb6 K69A and S70N are constitutive-active and constitutive-inactive mutants, respectively (Hunn et al, 2008). When wild-type (WT) MEFs were overexpressed with WT-, K69A-, or S70N-Irgb6, WT- and K69A-Irgb6 but not S70N-Irgb6 was detected on *T. gondii* PVM (Fig S4A and B), as described previously (Hunn et al, 2008). When Irgb6-deficient MEFs were reconstituted with WT-, K69A-, or S70N-Irgb6, only WT-Irgb6 was detected on *T. gondii* PVM in response to IFN-γ (Fig S4C and D). However, the K69A-Irgb6 mutant in WT cells was found to localize to *T. gondii* PVM, as described previously (Hunn et al, 2008). This suggests that this discrepancy may be related to the presence or absence of endogenous Irgb6 expression in WT or Irgb6-deficient cells, respectively. In WT cells, K69A-Irgb6 is likely to be dimerized with endogenous Irgb6 at *T. gondii* PVM.

In addition, IFN-γ–induced *T. gondii* killing activity was only rescued when WT-Irgb6 was overexpressed in Irgb6-deficient MEFs (Fig S4E). Conversely, expressing K69A- or S70N-Irgb6 in Irgb6-deficient cells failed to rescue the parasite killing mechanism (Fig S4E), indicating that GTPase activity is essential for the Irgb6-mediated cell-autonomous response against *T. gondii*.

### Irgb6 binds to PI5P and phosphatidylserine (PS), which are both detected at *T. gondii* PVM

The IRG Irgb6 belongs to the dynamin GTPase superfamily. These GTPases play an important role in the fission of clathrin-coated pits from the plasma membrane during endocytosis (Martens & Howard, 2006; Antonny et al, 2016). It has been determined that dynamin binding to acidic phospholipids is essential for dynamin-mediated membrane recognition (Salim et al, 1996; Lemmon & Ferguson, 2000). We hypothesized that acidic phospholipid binding would play a role in initial Irgb6-mediated *T. gondii* PVM recognition. We, therefore, performed a protein–lipid overlay assay using a His-tagged recombinant Irgb6 protein (Fig 4A), which revealed that Irgb6 mainly bound to monophosphorylated phosphoinositides (PIs) such as PI3P, PI4P, PI5P, or PS. Among these, the strongest interaction was observed with PI5P (Fig 4A). We next tested whether *T. gondii* PVM is composed of these phospholipids. Indeed, immunofluorescence staining revealed that anti-PS or anti-PIP2, which recognize PI3P, PI4P, PI5P, PI(3,5)P$_2$, PI(3,4,5)P$_3$, and PA (Fig S5A), stained *T. gondii* PVM (Fig 4B and C). This implies that PS or PIs may localize at *T. gondii* PVM. To visualize cellular PI3P, PI4P, PI5P, PI(3,4,5)P$_3$, and PS, we used HA-tagged p40-PX, OSBP, ING2-PHD, Btk-PH, and the MFG-E8-C2 domain, as previously described (Kanai et al, 2001; Hanayama et al, 2002; Levine & Munro, 2002; Gozani et al, 2003; Ebner et al, 2017). Notably, HA-tagged ING2-PHD and the MFG-E8-C2 domain were detected on the PVM, whereas HA-tagged p40-PX, OSBP, and Btk-PH were not (Fig 4D). Quantification analysis of immunofluorescence staining showed that about 34.0 or 9.7% of *T. gondii* vacuoles are HA-tagged ING2-PHD–positive or HA-tagged MFG-E8-C2 domain-positive, respectively. However, HA-tagged p40-PX, OSBP, and Btk-PH were detected much less frequently on the PVM (Fig 4E), suggesting that PI5P and PS can be recognized by Irgb6 at the PVM.

### Basic amino acid residues in C-terminal α-helices are important for both Irgb6 loading onto *T. gondii* PVM and phospholipid binding

We next sought to identify the Irgb6 regions involved in phospholipid binding–dependent *T. gondii* PVM recognition. The C-terminal α-K helix of regulatory IRG proteins seems to be crucial for the specificity of intracellular organelle targeting (Martens et al, 2004; Martens & Howard, 2006; Henry et al, 2014) or for targeting to the mycobacterial phagosome (Tiwari et al, 2009). A recent study showed that the C-terminal α-helical domains (especially αF and αK) of Irgb10 were predicted to be required for Irgb10 antimicrobial action involving intracellular bacterial membrane targeting (Man et al, 2016). Irgb6 is predicted to possess two C-terminal α-helical domains (αF and αK) based on the crystal structure analysis of Irga6, which provides the general structure for mouse IRG proteins (Fig 5A) (Ghosh et al, 2004). We, therefore, focused on two C-terminal α-helical domains (αF and αK) in Irgb6. When we deleted the αF and/or αK domain from WT Irgb6 and expressed them in Irgb6-deficient MEFs (Fig 5B and C), the mutants lacking α-helices were not loaded onto the PVM (Fig 5B and C). This suggests that the α-helical regions are important for Irgb6 targeting to *T. gondii* PVM. Given that both of the α-helices are amphipathic (Fig 5D) (Man et al,

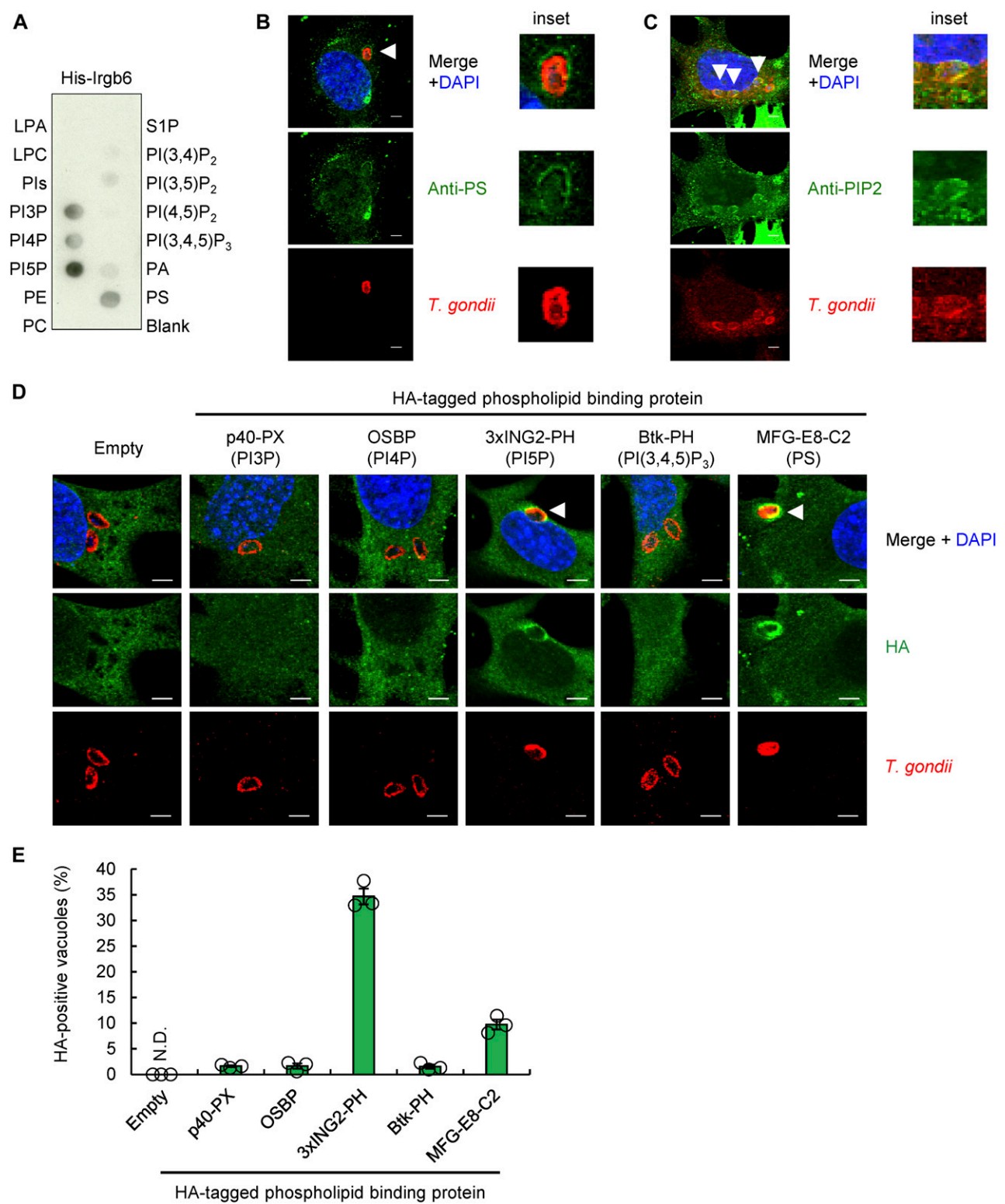

**Figure 4. Irgb6 recognizes PI5P and PS at *T. gondii* PVM.**
**(A)** The representative image from three independent experiments of PIP strips showing the binding of His-tagged recombinant Irgb6 protein to PI3P, PI4P, PI5P, and PS.
**(B)** Confocal microscope images from two independent experiments of the localization of PS (green) with *T. gondii* (red), and DAPI (blue) in WT MEFs. Scale bars, 5 μm.
**(C)** Confocal microscope images from three independent experiments of the localization of PIs (green) with *T. gondii* (red), and DAPI (blue) in WT MEFs. Scale bars, 5 μm.
**(D, E)** Confocal microscope images (D) and the graphs (E) from three independent experiments represent the localization of the indicated HA-tagged probes recognizing each specific PIs (green) with *T. gondii* (red), and DAPI (blue) in WT MEFs. Scale bars, 5 μm. White arrowheads indicate colocalization.

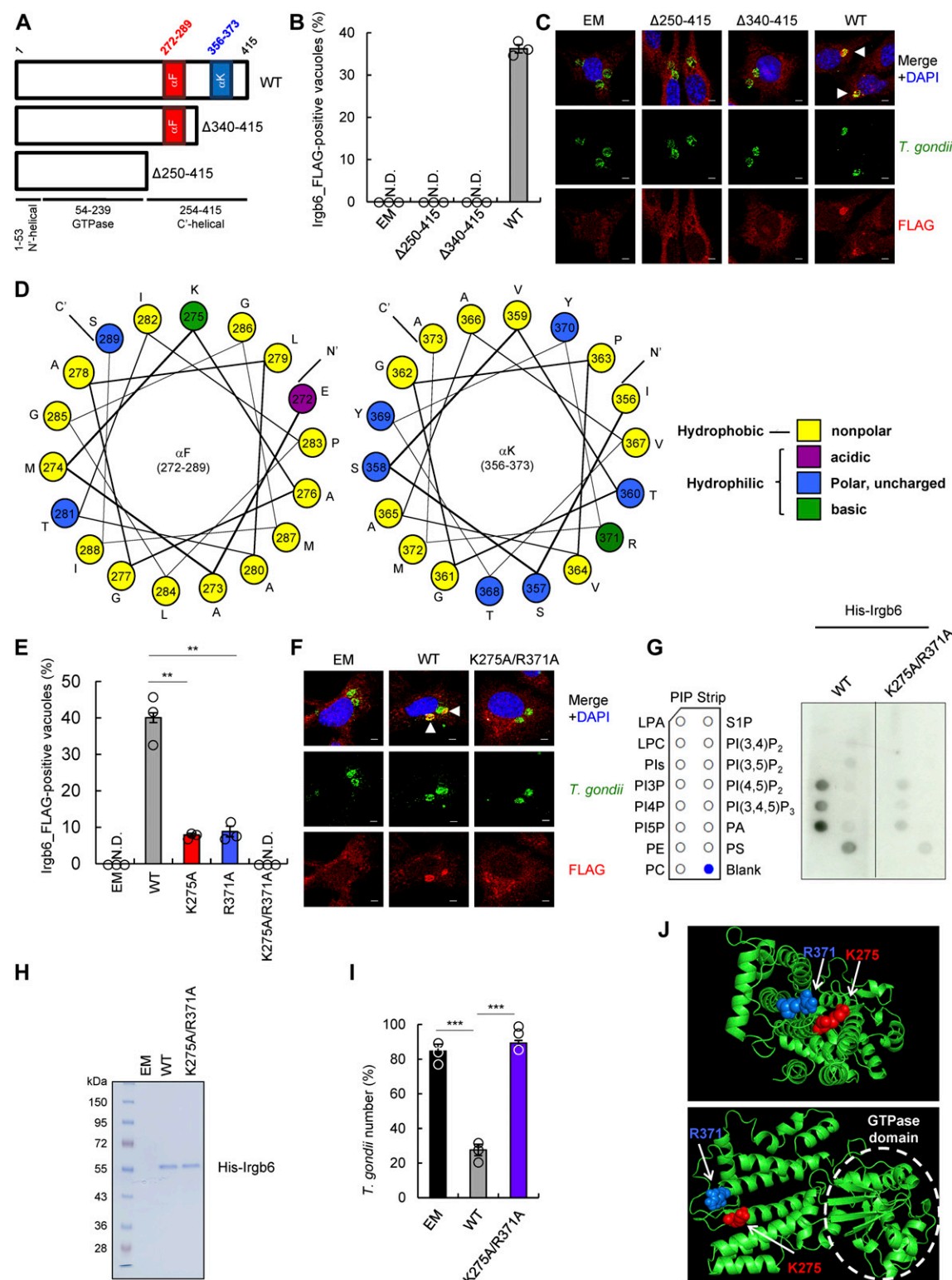

**Figure 5. Basic amino acids in the C-terminal α-helices of Irgb6 are required for recognition of phospholipids and immune responses against *T. gondii*.**
**(A)** Schematic representation of WT Irgb6 and C-terminal α-helices–deleted mutants of Irgb6. **(B, C)** The ratio (B) and representative images (C) of colocalization of FLAG-tagged WT Irgb6 or FLAG-tagged C-terminal deletion mutants Irgb6 (red) to *T. gondii* vacuoles (green) by confocal microscopy analysis at 4 h postinfection in Irgb6 KO MEFs reconstituted with the indicated proteins in the presence of IFN-γ stimulation. Scale bars, 5 μm. **(D)** The helical wheel projection of the two α-helical regions of Irgb6 and alignment according to its amphiphilic properties. Yellow indicates nonpolar amino acids, purple indicates acidic amino acids, blue indicates polar amino acids, and green indicates basic amino acids. **(E, F)** The ratio (E) and representative images (F) of colocalization of FLAG-tagged WT Irgb6 or FLAG-tagged the point mutants Irgb6 (red)

2016) and that PI5P and PS are negatively charged at physiological pH, we hypothesized that basic amino acid residues such as K275 and R371 (both of which are positively charged at physiological pH) are involved in phospholipid-mediated *T. gondii* PVM targeting. To test this, we generated Irgb6 mutants in which various basic amino acid residues around the $\alpha$-helical regions were replaced with alanine. We assessed whether the Irgb6 mutants were recruited to the PVM via ectopic expression in IFN-$\gamma$–stimulated Irgb6-deficient MEFs (Figs 5E and F, and S5B). We found that K275A and R371A accumulation on *T. gondii* PVM was severely impaired (Fig 5E). This did not occur for the K233A, K266A, K268A, or K395A Irgb6 mutants (Fig S5B), suggesting that K275 and R371 residues are specifically involved in Irgb6 loading on the PVM. Furthermore, recruitment was completely abolished in the Irgb6 mutant in which both K275 and R371 residues were replaced with alanine residues (K275A/R371A) (Fig 5E and F).

Next, we performed a protein–lipid overlay assay using the K275/R371 Irgb6 mutant protein. We found that the mutant was not able to bind to phospholipids compared with WT Irgb6 (Fig 5G). The signals were detected simultaneously to ensure equal exposure times. The efficiency of recombinant protein purification was probed by Coomassie blue staining, and both proteins were detected as a single bright band in each well of the gel corresponding to the expected product sizes (Fig 5H). Ectopic K275A/R371A expression in the Irgb6-deficient MEFs consistently failed to rescue IFN-$\gamma$–induced parasite killing activity (Fig 5I).

K275 and R371 residues were mapped onto the in silico Irgb6 structural model generated using the crystalized Irga6 structure (Ghosh et al, 2004). These basic amino acids were predicted to be on the very edge of the $\alpha$-helices, located on the opposite side of the GTPase domain and facing outside of the possible membrane-targeting region (Fig 5J). Collectively, these results show that basic amino acid residues in the C-terminal $\alpha$-helices of Irgb6 are important for initial targeting of phospholipid-mediated Irgb6 to *T. gondii* PVM and cell-autonomous immunity.

### Irgb6 provides in vivo host protection from *T. gondii* infection

The physiological relevance of Irgb6 in the host defense response against pathogens is unknown. We, therefore, investigated the role of Irgb6 in the host defense against *T. gondii* infection in vivo. To do this, we infected WT and Irgb6-deficient mice with luciferase-expressing type II Pru *T. gondii* and monitored their infection susceptibility profiles. Although the luciferase signals were comparably emitted from both WT and Irgb6-deficient mice on day 3 postinfection, the signals from the Irgb6-deficient mice on days 5 and 6 postinfection were greatly enhanced in comparison with those from the WT mice (Fig 6A and B). On day 5 postinfection, parasite numbers were counted in the peritoneal fluids or tissues

collected from the infected animals. It was found that Irgb6-deficient mice had higher parasite numbers than WT mice (Fig 6C). Moreover, the proinflammatory cytokine (e.g., IL-6, IL-12, and IFN-$\gamma$) levels in the peritoneal fluids from the *T. gondii*–infected Irgb6-deficient mice were enhanced compared with those in WT mice (Fig 6D). Furthermore, all Irgb6-deficient mice infected with *T. gondii* died within 9 d of infection, whereas all WT mice survived (Fig 6E). Thus, these results show that robustly increased parasite burden in Irgb6-deficient mice causes much higher proinflammatory responses and enhanced host susceptibility. This suggests that Irgb6 has a crucial role in initial binding on the PVM during the host defense response against *T. gondii* infection.

## Discussion

Soon after *T. gondii* invades IFN-$\gamma$–stimulated mouse cells, the parasite PVM becomes coated with multiple effector IRG proteins and is eventually disrupted (Martens et al, 2005). The loading process of effector IRG proteins is highly ordered, and Irgb6 and Irgb10 are reported to be the first in this process (Khaminets et al, 2010).

Here, we showed that Irgb6, but not Irgb10, plays a major role in anti–*T. gondii* cell-autonomous immunity. Irgb6-deficient cells were severely defective in the recruitment of ubiquitin and other IFN-$\gamma$–inducible GTPase effectors onto the *T. gondii* PVM. However, Irga6 or Irgb10 deficiency did not affect Irgb6 and ubiquitin coating on the PVM, indicating that Irgb6 has a larger role among effector IRG proteins. IFN-$\gamma$–dependent ubiquitination on *T. gondii* PVM is well established, ultimately resulting in parasite growth restriction via several mechanisms. These include mediating parasite vacuole–lysosome fusion and recruiting autophagy adaptors in human cells (Haldar et al, 2015; Selleck et al, 2015; Clough et al, 2016). Furthermore, the binding of ubiquitin and p62 to the PVM mediates parasite antigen-specific CD8$^+$ T cell activation in mice (Lee et al, 2015). However, the PVM substrate is not clear. Our previous study also showed that ubiquitin localization on the PVM is normal in MEFs deficient in chromosome 3 GBP family or Irga6, excluding the possibility that these are PVM substrates for ubiquitin (Lee et al, 2015). Interestingly, we found that ubiquitin localization on *T. gondii* PVM is dependent on Irgb6, but not Irgb10. These data, therefore, indicate the predominant role of Irgb6 in IFN-$\gamma$–induced cell-autonomous resistance against avirulent *T. gondii* type II. This occurs via enhancing the targeting of ubiquitin and other effector proteins to the PVM. Future study is required to clarify whether the Irgb6 is ubiquitinated on the PVM after targeting, or whether other proteins on the PVM are ubiquitinated after Irgb6-dependent PVM disruption.

to *T. gondii* vacuoles (green) by confocal microscopy analysis at 4 h postinfection in Irgb6 KO MEFs reconstituted with the indicated proteins in the presence of IFN-$\gamma$ stimulation. Scale bars, 5 $\mu$m. **(G)** The representative images from two independent experiments of PIP strip membranes incubated with His-tagged recombinant WT-Irgb6 or K275A/R371A-Irgb6 protein. **(H)** Coomassie blue staining of the purified His-tagged recombinant WT-Irgb6 and K275A/R371A-Irgb6 protein. **(I)** *T. gondii* survival rate in the presence of IFN-$\gamma$ stimulation relative to that in the non–IFN-$\gamma$–treated control by luciferase analysis at 24 h postinfection in Irgb6 KO MEFs reconstituted with empty (EM), WT-Irgb6, or K275A/R371A-Irgb6. **(J)** 3D models of the distribution of K275 (red) and R371 (blue) in Irgb6. **(B, E, I)** All graph shows the mean ± SEM in three independent experiments. **(C, F, G, H)** All images are representative of two or three independent experiments. **(C, F)** White arrowheads indicate colocalization. \*\*$P < 0.01$ and \*\*\*$P < 0.001$ from the two-tailed *t* test. ND, not detected.

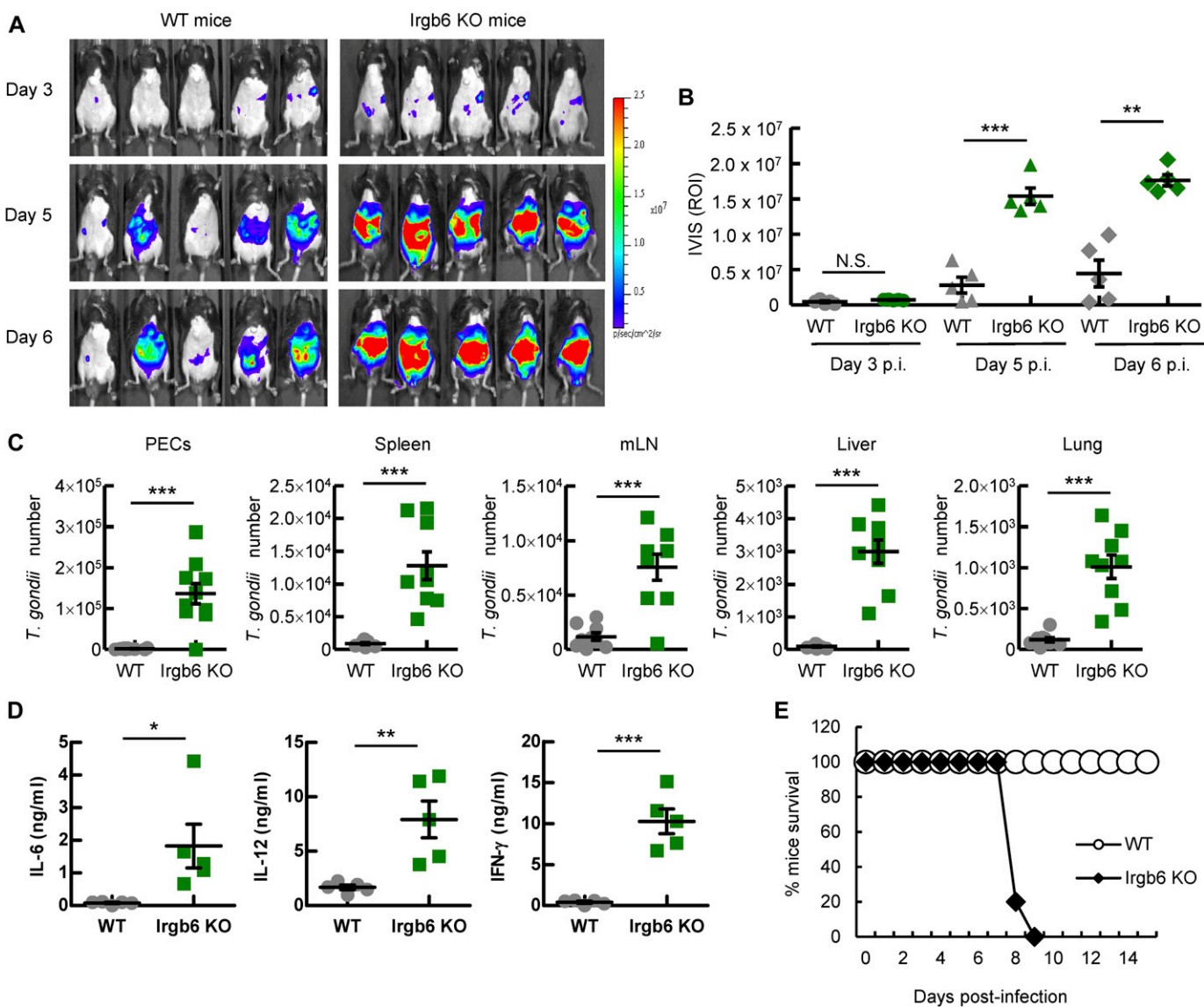

**Figure 6. Irgb6 provides host defense against *T. gondii* infection in vivo.**
**(A, B)** In vivo bioluminescence imaging (A) and quantification of *T. gondii* numbers (B) in WT and Irgb6 KO mice on day 3, 5, and 6 postinfection. **(C)** Parasite numbers in the indicated tissues of the mice by luciferase analysis on day 5 postinfection. **(D)** Release of proinflammatory cytokines in the peritoneal fluids of *T. gondii*–infected WT and Irgb6 KO mice on day 5 postinfection. **(E)** Survival rates of *T. gondii*–infected WT and Irgb6 KO mice. All images are representative of 5–10 mice per group, and all graphs are means ± SEM from two independent experiments. Two-tailed *t* tests were used. NS, not significant; *$P < 0.05$, **$P < 0.01$, and ***$P < 0.001$ versus WT mice.

We demonstrated that Irgb6 binding to the PVM is completely dependent on regulator IRG proteins such as Irgm1 and Irgm3. Furthermore, the universally conserved phosphate-binding loop GKS motif in the Irgb6 GTPase domain is essential for targeting to *T. gondii* vacuoles. Reconstituting Irgb6 S70N and K69A mutants in Irgb6-deficient cells revealed the essential role of GTPase activity when Irgb6 accumulates on *T. gondii* PVM, as shown by a previous study (Hunn et al, 2008).

We have also shown that the phospholipid binding activity of Irgb6 is required for PVM targeting. Recombinant Irgb6 protein was able to bind PS and PIs such as PI3P, PI4P, and PI5P in vitro. Cellular PI5P and PS also appeared to specifically accumulate at the PVM via unknown mechanisms. We found that K275 and R371 in the

C-terminal α-helical domains were the critical amino acid residues in Irgb6 for phospholipid binding and *T. gondii* PVM loading. Interactions between amphipathic α-helices of antimicrobial peptides and lipids are involved in targeting antimicrobial peptides to microbial membranes composed of negatively charged lipids (Dathe & Wieprecht, 1999; Shai, 1999; Mihajlovic & Lazaridis, 2010; Zhang & Gallo, 2016). Considering this, Irgb6 likely accumulates on the *T. gondii* PVM via an electrostatic interaction between K275/R371 (basic or positively charged at physiological pH) and PI5P/PS (acidic or negatively charged). Among the PIs, the role of PI5P in mammalian cells remains poorly understood (De Craene et al, 2017). Cellular PI5P levels are extremely low in many cell types relative to some other PIs (1–2% of PI4P) (Grainger et al, 2012; Viaud et al, 2014). In host cells, PI5P

can be altered by *Shigella flexneri* virulence factor IpgD. This converts PI(4,5)$P_2$ to PI5P, leading to direct actin remodeling and modification of the microbe uptake machinery (Niebuhr et al, 2002; Ramel et al, 2011; Boal et al, 2016; Hasegawa et al, 2017). Despite its presence in low levels, PI5P seems to be present in the plasma membrane (Sarkes & Rameh, 2010) and is phosphorylated by type II PIP4K to become PI(4,5)$P_2$ (Rameh et al, 1997). This is hydrolyzed into the $Ca^{2+}$-mobilizing messenger inositol trisphosphate and the protein kinase C activator diacylglycerol at the plasma membrane (Berridge & Irvine, 1984). PI5P also seems to be present in intracellular membrane compartments, including endosomes, the ER, and the Golgi apparatus (Sarkes & Rameh, 2010). Phosphatidylserine is synthesized in the ER and delivered to the plasma membrane by vesicular trafficking via the Golgi apparatus (Fairn et al, 2011). Although Irgb6 possibly recognizes these endomembranes, Irgm1 and Irgm3 may keep Irgb6 in a GDP-bound state, and hence an inactive form, to protect these organelles. Cytosolic endomembranous organelles such as the Golgi apparatus, mitochondria, ER, and endolysosomes are coated with regulator IRG proteins such as Irgm1 and Irgm3 (Hunn et al, 2008; Springer et al, 2013). Nevertheless, the protective mechanism preventing Irgb6 loading at the plasma membrane remains unclear. When *T. gondii* actively invades host cells, the moving junction, which forms at the host cell entry site, serves as a molecular sieve for host proteins. These protect the parasite PVM from lysosomal degradation (Mordue et al, 1999). The molecular sieve formed by moving junction formation during *T. gondii* invasion may alter PI5P levels at the host plasma membrane, possibly increasing PI5P levels on the PVM and triggering Irgb6 loading. In terms of phospholipids and the IRG protein, Irgm1 was previously shown to interact with PI(3,4)$P_2$ and PI(3,4,5)$P_3$ and is important for localization to bacteria-containing phagosomes (Tiwari et al, 2009). However, whether Irgm1 binding to phospholipids is important for IFN-γ–mediated immunity to *T. gondii* remains unclear. Whether Irgm1 localizes to bacteria-containing phagosomes is currently disputed (Springer et al, 2013).

Finally, our findings demonstrated the protective role of Irgb6 in vivo against *T. gondii* infection. Irgb6 deficiency causes markedly increased parasite burden in the tissues and increased inflammatory responses. This suggests that upon Irgb6-deficiency, failure of PVM disruption during early *T. gondii* infection eventually leads to host death. During in vivo infection, *T. gondii* can spread to other parts of the body via interaction with immune cells, such as neutrophils and inflammatory monocytes (Lambert et al, 2006; Bierly et al, 2008; Unno et al, 2008; Dunay et al, 2010; Coombes et al, 2013). Consistently, we found that Irgb6-deficient mice displayed significantly higher numbers of *T. gondii*–infected inflammatory monocytes and neutrophils in the peritoneal cavity compared with wild-type mice (data not shown).

In summary, our studies discuss the fact that not much is known about the mechanisms by which IRG proteins recognize and destroy *T. gondii* PV and proceed to investigate this in the context of Irgb6. Our findings highlight an active host defense program relying on the initial binding of phospholipid-dependent Irgb6 to *T. gondii* PVM. The initial binding of Irgb6 on *T. gondii* vacuoles is likely to involve PI5P and PS and is a crucial feature of ubiquitination, PVM destruction, and synergistical *T. gondii* killing activity. Our studies cover topics involving Irgb6 gene knockout, host–pathogen interactions, and immune responses. This study, therefore, appropriately

highlights its contribution to existing research. It will be necessary to examine whether other IFN-inducible GTPases targeting bacterial and fungal membranes recognize the same or different phospholipids and induce the membrane remodeling. This could further reveal the biological significance of IFN-inducible GTPase-dependent cell-autonomous immunity.

# Materials and Methods

## Reagents

Antibodies against Irgb6 (TGTP; sc-11079), GBP1-5 (sc-166960), Irgm1 (LRG-47; sc-11075), Irgm2 (GTPI; sc-11088), Irgm3 (IGTP; sc-136317), and His-probe (H-3; sc-8036) were purchased from Santa Cruz Biotechnology, Inc. Antibodies against FLAG M2 (F3165) and β-actin (A1978) were purchased from Sigma-Aldrich. Mouse monoclonal and rabbit polyclonal anti-HA antibodies were obtained from BioLegend and Sigma-Aldrich, respectively. Rabbit polyclonal anti-GBP2 and mouse monoclonal anti-p62 (PM045) antibodies were obtained from Proteintech and MBL International, respectively. Anti-ubiquitin mouse monoclonal antibody (FK2; MFK-004) was obtained from Nippon Biotest Laboratories. Mouse monoclonal anti-Irga6 (10D7) and rabbit polyclonal anti-Irgb10 antibodies were provided by Dr JC Howard (Instituto Gulbenkian de Ciência). Rabbit polyclonal anti-GBP1 antibody was provided by Dr EM Frickel (Francis Crick Institute). Rabbit polyclonal anti-GRA7 antibody was provided by Dr JC Boothroyd (Stanford University School of Medicine). Mouse monoclonal anti-GRA2 and rabbit polyclonal anti-Gap45 antibodies were provided by Dr D Soldati-Favre (University of Geneva). Anti-PIP2 antibody (2C11) was purchased from Abcam. Anti-PS antibody (1H6) was purchased from Millipore. Recombinant mouse IFN-γ was purchased from PeproTech.

## Mice

All animal experiments were approved by the Animal Research Committee of the Research Institute for Microbial Diseases (Osaka University, Osaka, Japan). Irgb6-deficient C57BL/6 mice and their wild-type counterparts were maintained under specific pathogen-free conditions and used for experimental study at 8–10 wks. Irgb10-deficient mice were constructed as previously described (Man et al, 2016).

## Cell culture

Primary MEFs were maintained in DMEM (Nacalai Tesque) supplemented with 10% heat-inactivated FBS (JRH Bioscience), 100 U/ml penicillin (Nacalai Tesque), and 100 μg/ml streptomycin (Nacalai Tesque). Bone marrow–derived macrophages were generated by cultivating BM progenitors isolated from the BM in complete medium, containing 10% L-cell conditioned medium, for 6–7 d. The complete medium consisted of 10% heat-inactivated FBS, 10 mM Hepes (Sigma-Aldrich), 100 U/ml penicillin, 100 μg/ml streptomycin, 50 μg/ml gentamicin (Sigma-Aldrich), 10 μg/ml polymyxin B (Sigma-Aldrich), 1 mM sodium pyruvate (Sigma-Aldrich), 50 μM 2-mercaptoethanol

(Gibco), 1 mM nonessential amino acids (Gibco), and 2 mM L-glutamine (Gibco) in RPMI1640 medium (Nacalai Tesque).

### Generation of MEFs derived from Irgb6-deficient, Irgb10-deficient, or Irgb6/b10 double-deficient mice by Cas9/CRISPR genome editing

The insert fragment of Irgb6 gRNA was amplified using KOD FX NEO (Toyobo) and the following primers: Irgb6_gRNA1_F, Irgb6_gRNA1_R, Irgb6_gRNA2_F, and Irgb6_gRNA2_R. The fragment for gRNA was inserted into the gRNA-cloning vector (Plasmid 41824) using Gibson Assembly mix (New England Biolab) to generate gRNA-expressing plasmids. The T7 promoter was added to the gRNA template using KOD FX NEO and *Irgb6*_T7gRNA_F and gRNA_common_R primers. The T7-*Irgb6* gRNA PCR product was gel purified and used as the subsequent generation of gRNA. MEGAshortscript T7 (Life Technologies) was used for gRNA generation. Cas9 mRNA was generated by in vitro transcription using mMESSAGE mMACHINE T7 ULTRA kit (Life Technologies). The template was amplified by PCR using pEF6-hCas9-Puro and T7Cas9_IVT_F and Cas9_R primers (Ohshima et al, 2014), followed by gel purification. Irgb10 gRNA was previously prepared (Man et al, 2016). The synthesized gRNA and Cas9 mRNA were purified using a MEGAclear kit (Life Technologies) and eluted in RNase-free water (Nacalai Tesque). To obtain Irgb6-deficient, Irgb10-deficient, or Irgb6/b10 double-deficient MEFs, C57BL/6 female mice (6 wks old) were superovulated and mated to C57BL/6 stud males. Fertilized one–cell stage embryos were collected from oviducts and injected into the pronucleus or cytoplasm with Cas9 mRNA (100 ng/μl) and Irgb6 and/or Irgb10 gRNA (50 ng/μl). The injected live embryos were transferred into oviducts of pseudo-pregnant ICR females at 0.5 dpc. D13.5 embryos were collected to generate primary MEFs. The Irgb6 and Irgb10 protein expression by the resulting MEFs was analyzed by Western blot using antibodies against Irgb6 or Irgb10, respectively. Irga6-deficient MEFs were constructed as previously described (Liesenfeld et al, 2011). The male pup harboring the mutation was mated to C57BL/6 female mice and tested for germ line transmission. Heterozygotic mice were intercrossed to generate homozygotic Irgb6-deficient mice for in vitro and in vivo assays. The deleted region in Irgb6 genomic DNA was verified by sequencing analysis. The complete coding region of the Irgb5-b4 tandem decoy cDNA in Irgb6-deficient MEFs was confirmed by sequencing analysis.

### Type II *T. gondii* parasites

Parental PruΔHX and luciferase-expressing PruΔHX were maintained in Vero cells by passaging every 3 d in RPMI1640. This was supplemented with 2% heat-inactivated FBS, 100 U/ml penicillin, and 100 μg/ml streptomycin.

### Cloning and recombinant expression

The cDNA regions of interest corresponding to the wild-type control and indicating Irgb6 point or deletion mutations (GenBank accession no. NM_001145164) were synthesized from mRNA from the spleen of C57BL/6 mice. The cDNA used to generate the HA-tagged PHD of p40, OSBP, ING2, or Btk were obtained from Addgene. The

cDNA used to generate the HA-tagged MFG-E8-C2 were kindly provided by Dr S Nagata (Osaka University). The Irgb6 mutants were generated using primers (Table S1). The PCR products were then ligated into the retroviral pMRX expression or pEU-E01-MCS vector (CellFree Sciences) for retroviral infection or recombinant protein expression, respectively. The sequence of all constructs was confirmed by DNA sequencing.

### Generation of recombinant Irgb6 using cell-free system

His-tagged recombinant Irgb6 WT and K275A/R371A were expressed with the wheat germ cell-free system (CellFree Sciences) and purified with a Nickel-Sepharose six fast flow (GE Healthcare), as previously reported (Tsuboi et al, 2010). Protein purity was evaluated by SDS–PAGE and Coomassie brilliant blue staining.

### Immunofluorescence study

Cells were cultured on glass coverslips in the presence or absence of IFN-γ (10 ng/ml) for 24 h. Cells were infected with type II *T. gondii* (MOI of 5 or 1) for 4 h and fixed in PBS containing 3.7% paraformaldehyde for 10 min at room temperature. Cells were permeabilized with PBS containing 0.1% Triton X-100 or 0.002% digitonin for 5 min, followed by blocking with 8% FBS in PBS for 1 h at room temperature. Cells were subsequently incubated with the indicated primary antibodies for 1 h at 37°C, followed by incubation with Alexa 488-, Alexa 594-, or Alexa 647-conjugated secondary antibodies (molecular probes) and DAPI for 1 h at 37°C in the dark. Finally, coverslips were mounted onto glass slides with PermaFluor (Thermo Fisher Scientific) and analyzed using confocal laser microscopy (FV1200 IX-83; Olympus). All images are shown at 1,000× magnification. For PIP2 and PS staining, the cells were fixed in PBS containing 3.7% paraformaldehyde and 0.2% glutaraldehyde for 3 h at 4°C. After washing with PBS containing 50 mM NH₄Cl, cells were incubated in PBS containing 0.5% saponin, 8% FBS, and 50 mM NH₄Cl for 3 h at 4°C. The cells were then incubated overnight with primary antibodies in PBS containing 0.1% saponin and 8% FBS at 4°C. Following this, cells were incubated with Alexa 488- or Alexa 594-conjugated secondary antibodies and DAPI for 1 h at 4°C. All images are shown at 1,000× magnification.

### Transmission electron microscopy

Primary MEFs treated with 10 ng/ml IFN-γ for 24 h were infected with type II *T. gondii* (MOI of 5) for 4 h. After washing with PBS, the cells were fixed overnight with 2.5% glutaraldehyde in 0.1 M phosphate buffer at 4°C. The cells were post-fixed with 1% OsO₄ in the same buffer for 1 h at 4°C, dehydrated in a graded series of ethanol, and embedded in Quetol 812 (Nissin EM). Silver sections were cut with an ultramicrotome, stained with lead citrate and uranyl acetate, and observed with an H-7650 electron microscope (Hitachi).

### Western blot analysis

Cells were washed with PBS and lysed with 1× TNE buffer (20 mM Tris–HCl, 150 mM NaCl, 1 mM EDTA, and 1% NP-40) containing protease inhibitor cocktail (Nacalai Tesque). A total protein was

loaded onto and separated on a 10% or 15% SDS–PAGE and transferred to a polyvinyl difluoride membrane. The membrane was blocked with 5% skim milk in PBS/Tween 20 (0.2%) and probed overnight with the indicated primary antibodies at 4°C. After washing with PBS/Tween, the blot was probed with HRP-conjugated secondary antibodies for 1 h at 25°C and visualized by Luminata Forte Western HRP substrate (Millipore).

### Protein–lipid overlay assay

A protein–lipid overlay assay was performed with 0.5 µg/ml of His-tagged recombinant proteins using PIP strips (Echelon Biosciences), as per the manufacturer's instructions. Lipid binding was immunodetected with a combination of anti-His tag mouse monoclonal antibody and HRP-conjugated rabbit–anti-mouse IgG, followed by Luminatz Forte Western HRP substrate. In detail, the PIP strip membrane (P-6001; Echelon Biosciences Inc.) was blocked with TBS-T (0.1% Tween-20) + 3% BSA (fatty acid free) for 1 h at 25°C with gentle agitation. Next, we discarded blocking buffer and added 0.5 µg/ml purified recombinant protein Irgb6-His tagged in 1.5 ml TBS-T + 3% BSA and incubated overnight at 4°C. Next, we discarded the protein solution and washed with TBS-T five times with gentle agitation for 5 min each. Next, we discarded wash buffer and added 1.5 µl anti–His-HRP antibody in 1.5 ml TBS-T + 3% BSA and incubated the membrane for 1 h at 25°C, with gentle agitation. Next, we discarded the antibody solution and washed with TBS-T six times with gentle agitation for 10 min each. Next, we discarded wash buffer and detected the bound protein by ECL development followed by ImageQuant LAS 4000 (GE Healthcare). The displayed images in Figs 4 and 5 were used from ImageQuant LAS 4000.

### Mice survival and in vivo measurement of parasites by imaging

Mice were intraperitoneally infected with PruΔHX *T. gondii* tachyzoites expressing luciferase ($1 \times 10^4$ in 200 µl PBS per mouse), and the survival of the mice was monitored for 15 d postinfection. For in vivo imaging of the parasites, mice were intraperitoneally injected with 3 mg D-luciferin in 200 µl PBS (Promega) on days 3, 5, and 6 postinfection. After inhalational anesthesia with isoflurane (Sumitomo Dainippon Pharma), abdominal photon emission was assessed during 60-s exposure by an in vivo imaging system (IVIS-Spectrum; Xenogen), followed by analysis with Living Image software (Xenogen).

### Luciferase assay

To measure the number of *T. gondii*, cells were left untreated or treated with IFN-γ (10 ng/ml) for 24 h. Following this, cells were infected with luciferase-expressing PruΔHX *T. gondii* (MOI of 1) for 24 h. The infected cells were harvested and lysed with 100 µl 1× passive lysis buffer (Promega) with sonication for 30 s. To measure the number of *T. gondii* in the tissues, peritoneal cavity, mesenteric lymph nodes, spleen, liver, and lungs, these sections were removed on day 5 postinfection. The mesenteric lymph nodes, spleen, liver, and lungs were homogenized and lysed in 1 ml 1× passive lysis buffer with sonication. After centrifugation at 13,000*g* for 10 min at 4°C, luciferase activity was measured using 5 µl of the supernatants

via the dual-luciferase reporter assay system (Promega) by GLOMAX 20/20 luminometer (Promega). The in vitro data are presented as the percentage of *T. gondii* survival in IFN-γ–stimulated cells relative to unstimulated cells (Con). The in vivo data are presented as absolute values.

### ELISA

The level of mouse IL-6, IL-12, and IFN-γ secretion was measured by ELISA analysis according to the manufacturer's protocol (eBioscience).

### In silico structural modeling

For structural modeling of Irgb6, the Irga6 structure (Protein Data Bank ID 1TQ4) was used as a template with HHpred alignments and the spanner three-dimensional rendering tool. Figures were prepared using the PyMOL molecular graphics system (version 1.7.6.0, Schrödinger, LLC).

### Statistical analysis

Statistical analyses were performed using the *t* test. *P*-values less than 0.05 were considered statistically significant. All graphs show the mean ± SEM of three independent experiments (three biological replicates).

## Supplementary Information

## Acknowledgements

We thank Mari Enomoto (Osaka University) for secretarial and technical assistance. We thank Drs Jonathan C Howard, Eva-Maria Frickel and Dominique Soldati-Favre for providing anti-Irgb10, Irga6, Gbp1, and GRA2 antibodies. We thank Dr Shigekazu Nagata for providing MFG-E8 expression vector. This study was supported by the Research Program on Emerging and Re-emerging Infectious Diseases (JP19fk0108047) and Japanese Initiative for Progress of Research on Infectious Diseases for global Epidemic (JP19fm0208018) from the Agency for Medical Research and Development (AMED), Grant-in-aid for scientific research on innovative areas (production, function, and structure of neo-self; 19H04809), for scientific research (B) (18KK0226 and 18H02642) and for scientific research (A) (19H00970) from the Ministry of Education, Culture, Sports, Science and Technology, Cooperative Research Grant of the Institute for Enzyme Research, Joint Usage/Research Center, Tokushima University, Takeda Science Foundation, Mochida Memorial Foundation on Medical and Pharmaceutical Research, Uehara Memorial Foundation, Naito Foundation, Astellas Foundation for Research on Metabolic Disorders, and Research Foundation for Microbial Diseases of Osaka University.

### Author Contributions

Y Lee: conceptualization, resources, data curation, software, formal analysis, funding acquisition, validation, investigation, visualization, and writing—original draft, review, and editing.

H Yamada: data curation, visualization, and methodology.

A Pradipta: data curation and formal analysis.

JS Ma: data curation and formal analysis.

M Okamoto: investigation.

H Nagaoka: resources and investigation.

E Takashima: resources and investigation.

DM Standley: software, investigation, and visualization.

M Sasai: conceptualization and investigation.

K Takei: conceptualization and resources.

M Yamamoto: conceptualization, resources, data curation, formal analysis, supervision, funding acquisition, validation, investigation, visualization, methodology, project administration, and writing—original draft, review, and editing.

## Conflict of Interest Statement

The authors declare that they have no conflict of interest.

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
