## [Reviewer comments · Life Science Alliance]

Life Science Alliance

Initial phospholipid-dependent Irgb6 targeting to *Toxoplasma gondii* vacuoles mediates host defense

Youngae Lee, Hiroshi Yamada, Ariel Pradipta, Ji Ma, Masaaki Okamoto, Hlkaru Nagaoka, Eizo Takashima, Daron Standley, Miwa Sasai, Kohji Takei, and Masahiro Yamamoto

DOI: <https://doi.org/10.26508/lsa.201900549>

Corresponding author(s): Masahiro Yamamoto, Research Institute for Microbial Diseases

Review Timeline:

Submission Date:	2019-09-11
Editorial Decision:	2019-09-24
Revision Received:	2019-11-25
Editorial Decision:	2019-11-28
Revision Received:	2019-12-02
Accepted:	2019-12-02

Scientific Editor: Andrea Leibfried

Transaction Report:

September 24, 2019

Re: Life Science Alliance manuscript #LSA-2019-00549-T

Prof. Masahiro Yamamoto
Graduate School of Medicine, Osaka University
Osaka University
2-2 Yamadaoka
Suita city, Osaka 565-0871
Japan

Dear Dr. Yamamoto,

Thank you for submitting your manuscript entitled "Phospholipid-dependent Irgb6 initial targeting to Toxoplasma gondii vacuoles mediates host defense" to Life Science Alliance. The manuscript was assessed by expert reviewers, whose comments are appended to this letter.

As you will see, the reviewers think that your work is of value to the field, pending satisfactory major revision. However, quite an extensive revision is needed. Reviewer #2 provides constructive input on how to strengthen your work by adding appropriate controls (KO generation; comparison to Irgm1/Irgm3 double deficient condition), changing the data representation and by considering the value of the tubulation and lipid vesicle disruption analyses in light of the fact that the effects are independent of nucleotide. Both reviewers also point out that major efforts are needed for re-writing the manuscript and reviewer #2 points you to the most critical parts. We would thus like to invite you to submit a revised version of your work, addressing the criticisms raised.

Thank you for this interesting contribution to Life Science Alliance. We are looking forward to

receiving your revised manuscript.

Sincerely,

B. MANUSCRIPT ORGANIZATION AND FORMATTING:

Reviewer #1 (Comments to the Authors (Required)):

The manuscript by Lee et al addresses the role of *Irgb6* in resistance to *Toxoplasma gondii*. Prior work has shown that *Irgb6* localizes to the *T. gondii* vacuole (PV) in cells, and that it does so early in the cascade of IRG proteins that assemble on the vacuole. The current work advances our concept of *Irgb6* function in several important respects. The reviewers show that deletion of *Irgb6* greatly impairs immune attack on the *T. gondii* PV in contrast to lesser roles for other GKS proteins, indicating that *Irgb6* is not only a sentinel but a pivotal and essential factor in the IRG/GBP cell autonomous *T. gondii* resistance program. The investigators show that *Irgb6* is able itself to drive tabulation/vesiculation of membrane vesicles. Further, *Irgb6* seems to target PI5P-rich membranes, and it does so through basic residues in its amino terminus. Together, this assembles many aspects of the mechanism through which *Irgb6* functions.

This work will make an important addition to the literature. The series of studies are logically designed, the manuscript well-constructed, and statistics applied appropriately. The main deficiency in the manuscript is that many sentences are poorly constructed grammatically, in some cases altering their meaning. Careful editing should be performed to improve the grammar and sentence structure.

Reviewer #2 (Comments to the Authors (Required)):

Brief Summary: The central advance in this paper is a CRISPR-Cas9 KO of the duplicated *Irgb6* gene on Chromosome 11 of the mouse and the discovery and analysis of a strong disease susceptibility phenotype. *Irgb6* is a member of the multigene Immunity-Related GTPase (IRG) protein family in the mouse that is required for resistance to a small group of intracellular pathogens including *Toxoplasma gondii*. They are subdivided into 3 functional classes, effectors, regulators and decoys, each with distinctive roles in the overall resistance mechanism, which is associated with breakdown of the parasitophorous vacuole membrane (PVM). *Irgb6* is an effector. Until now, 2 effector IRG genes have been knocked out, *Irga6* and *Irgd*, both with significant but small effects on resistance against *Toxoplasma gondii*, and neither KO has a large effect on the loading of the other effector IRG proteins onto the PVM. The KO of *Irgb6* proves to have a large effect on resistance, the other effector IRG proteins reach the PVM inefficiently and the PVM remains intact. It was reported several years ago that the loading of IRG proteins onto the PVM is a structured process, with *Irgb6* as the first loaded "pioneer". The present results indicate that indeed *Irgb6* is the pioneer, and that this status is not just first in a time series but also required for the normal loading of the other effectors.

The lack of injury to the PVM in the *Irgb6* KO unsurprisingly leads to failure of the downstream effects of the IRG protein mediated resistance mechanism, namely the arrival at the PV of the 65 kDa GBPs, ubiquitin and the p62 autophagy adaptor.

More novel are the experiments that appear to identify specificity for PI5P on the PVM as the target for *Irgb6*. This is the first time that the basis of the highly specific targeting of the largely cytosolic effector IRG proteins to the PVM has been analysed. However, in view of the fact that other membranes also express PI5P, the specificity of PVM targeting is not fully accounted for in this study and there is certainly room for more work here.

Addition of soluble *Irgb6* to giant unilamellar vesicles (GUVs) leads to distortion and vesicular breakdown, perhaps suggesting something of the mode of action in causing vacuolar breakdown.

Review:

General Comments.

The results in this paper are of considerable interest in the field and should be published. However, the study is absolutely central to my own interests and work, and I am therefore at risk of being overcritical, without, I hope, any conflicts of interest. I should also say that the work was reported at a recent meeting where I discussed it with the senior author. The editors should bear this in mind when considering my review.

The paper is not very well written, significantly unclear in places and the English is often faulty. It is important that it be read and edited by a "native English speaker" but it is always the clear and correct meaning of the science that must be the basis for language edits.

I have made a large number of detailed points below. Here I would like to highlight a few of these that I consider of greatest importance.

Introduction:

Nomenclature. The authors make correct use of the gene and protein names defined by Bekpen et al, but they continue to use the GKS and GMS nomenclatures which divide IRG proteins into two functional classes. We introduced this dichotomous nomenclature when we first characterized the IRG protein family (Bekpen et al, 2005), using it only as a shorthand for the different sequence families. Subsequently, we showed that the GMS subgroup have a unique and essential function as negative regulators for GKS proteins that bind to the PV and act as effectors. However, in this dichotomous system another structurally distinct group of IRG proteins, the "tandems" was ignored, or rather, included in the GKS group through having classical nucleotide binding sites. We have since shown that at least one of these tandem IRG proteins has yet a third function, neither effector nor regulator, but acts to distract kinases secreted by virulent strains of *T. gondii*, preventing them from phosphorylating and inactivating the effector IRG proteins. We now therefore call the tandem IRG genes "decoys" (Mueller and Howard 2016).

I therefore suggest that the authors stop using the rather clumsy GMS/GKS nomenclature, that carries no functional information, and prefer to use "effector IRG proteins" or "effectors" and "regulator IRG proteins" or "regulators" (and "decoys" but they don't have any of those in this paper).

Referencing. I have made a substantial number of proposals for referencing that I believe are more correct and relevant. I specifically urge that the authors check directly with the references themselves whether they can agree with these changes or additions.

Active or passive targeting of IRG proteins to the PV. The authors confuse the process by which the IRG proteins reach the PV and their tendency to stay there once they have arrived. Only the latter is relevant to their observations on PI5P interactions. This must be sorted out.

Results:

Description and analysis of the Crispr.cas mutations. In view of the multiple uncertainties associated with the CRISPR-Cas9 method the authors must present specific sequence data on the nature of mutations they have introduced and at least state whether the *Irgb5-4* tandem (decoy) gene that lies between the two mutated *Irgb6* genes, is still present and unmutated. Can of course be in supplementary.

Loss of cell-autonomous resistance in *Irgb6*-deficient cells, Fig 1C. This figure shows the extent to which the resistance of IFN γ -induced cells is damaged in the *Irgb6* KO cells. The presentation is

misleading since the absence of resistance in cells not treated with IFN γ is only implied on the ordinate as "T. gondii numbers %". If untreated cells are 100%, then what Fig1 shows is that fully 70% of the IFN γ -dependent resistance is still present in the b6-deficient cells. This would be obvious if another column equal to 100% were introduced into the figure. As it stands, the impression given is that resistance is completely lost, but this is clearly not so. This is already confusing. However perhaps more important is that the authors themselves do not draw attention to the substantial residual resistance and do not comment on it. Is it due to action by other effector IRG proteins like Irga6, whose access to the PV is not completely inhibited in the Irgb6-deficient cells (Fig 2A,B). Or else what? In order to better assess the scale of the deficiency it would be interesting to compare the Irgb6 deficient cells with Irgm1/Irgm3 double-deficient cells, which lack two of the three regulators and are generally considered to be completely deficient in IRG-mediated resistance. I consider this an important issue.

PIP binding domain results Fig 4D. The single microscope images shown here are not adequate. These results must be quantitated for all 5 proteins.

Tubulation and disruption of lipid vesicles, Fig 5A-C. Very surprisingly, these effects of co-incubating lipid vesicles with the purified Irgb6 protein are independent of nucleotide. Since this is discordant with the cellular data that clearly show nucleotide-dependence, it is unclear what the observed effects are due to. In principle, this is an interesting experiment, but the lack of nucleotide dependence means that its relevance to the function of Irgb6 is not clear. I recommend eliminating this result from the paper, unless it can be supported by evidence for nucleotide-dependence. I note that no data is presented for characteristics such as the purity of the "purified" Irgb6 protein or its functional integrity, as would normally be provided in support of such a biophysical experiment.

Detailed comments:

Abstract:

The abstract is full of problems. I here suggest an improved text, but cannot commit to doing this much work for the whole paper

Suggested corrected text:

Toxoplasma gondii (*T. gondii*) is an obligate intracellular protozoan parasite capable of infecting all warm-blooded animals by ingestion. The organism enters host cells and resides in the cytoplasm in a membrane-bounded parasitophorous vacuole (PV). The induction of an interferon response enables IFN γ -inducible immunity-related GTPases (IRG proteins) to accumulate on the PV and restrict parasite growth. However, little is known about the mechanisms by which IRG proteins recognize and destroy the *T. gondii* PV. We here characterized the role of Irgb6, an IRG protein, in the cell-autonomous response against *T. gondii*, entailing the ubiquitination and breakdown of the vacuole. We show that Irgb6 is capable of binding a specific phospholipid present on the PV membrane, and that absence of Irgb6 results in reduced or absent targeting of other effector IRG proteins to the PV implying that Irgb6 has a role as pioneer in the complex process by which multiple IRG proteins access the PV. Irgb6 deficient mice are highly susceptible to infection with a strain of *T. gondii* avirulent in wild-type mice.

Notes:

The first two sentences unbundle the clumsy first sentence of the authors' abstract.

Parasites are NOT cleared by the IRG protein response; some parasites always survive and these are responsible for persistent infection in brain cysts. However the IRG protein attack undoubtedly restricts parasite growth.

Death of mice from a failure of the IRG system was shown many years ago to result from unrestrained growth of a normally avirulent *T. gondii* strain and high levels of inflammatory cytokines. I do not see the point of restating this here as it is not a new finding.

I find the last sentence of the authors' abstract is redundant.

Introduction

As everywhere, difficult text.

Line 55: I am unaware of any virus infection resisted by IRG proteins; references to the very important Chlamydia studies are missed completely.

Line 57: "IRGs consist of three regulatory etc etc" No; the IRG protein family in mice consists of.....

Line 57: There are many more genes than described proteins. The documented expressed IRG effector proteins are only 4: *Irga6*, *Irgb6*, *Irgb10* and *Irgd*. Better to write "There are 4 known expressed effector IRG proteins, *Irga6*, *Irgb6*, *Irgb10* and *Irgd*"

Line 61: GTPase activity has been demonstrated and published in detail only for *Irga6* and this should be referenced (Uthaiyah et al 2003, Hunn et al 2008). An early important, but not yet confirmed, report from Taylor and colleagues (JBC 271, 20399-20405) reported GTPase activity from *Irgm3* (IGTP), a "GMS" protein.

Lines 63 - 68 are hard to understand. The three GMS proteins are regulators that maintain the effectors ("GKS") proteins in an inactive GDP-bound state, probably to prevent the latter from activating inappropriately on host cell membrane-bounded vesicular systems. In their absence, effector proteins form what are probably GTP-bound aggregates and are unable to interact with the *T. gondii* PV. Correct references please.

Line 68,69: Is it really true that all 11 mouse GBPs have demonstrated GTPase activity?

Having an apparently intact nucleotide-binding site is not the same as having GTPase activity.

Line 72. Add "vesiculated" with disrupted, to anticipate data in Fig 1D

Line 72/73. Reference to Collazo et al is inappropriate here. Remove.

Lines 77-90 This paragraph confuses the mechanism by which IRG and GBP proteins initially encounter a PV, which at least in the case of IRG proteins, is almost certainly diffusion, and the mechanism that causes the IRG proteins to be retained at the vacuole, which is what is referred to in this paper.

Lines 90-101 appear to suggest that *Irgb6* is both necessary and sufficient for resistance against *T. gondii*. The experiments described in the paper show only that *Irgb6* is required for the full resistance response. Since both *Irga6* and *Irgd* have both also been shown to be required for wild-type levels of resistance it is clear that *Irgb6* may well not be sufficient. There is no evidence that a mouse possessing only *Irgb6* from the effector set could fully resist *T. gondii*. It would now be possible to prepare such a mouse and test the issue.

Results

Line 105 and elsewhere. Authors always use the word "clearance" to describe the cell-autonomous process of vacuole destruction, but the term is normally used to describe an animal free from infection, which is not the case for *T. gondii*. IFNg-induced resistance of cells to *T. gondii* is accompanied by necrotic death of the cells. Clearance is misleading and must be changed.

Line 106/107. It is wrong to refer to the susceptibility of *Irga6* KO mice as "controversial". The reference Liesenfeld et al 2011 describes the data obtained by two independent laboratories using

two independent KOs of *Irga6*. The survival data is essentially indistinguishable between the two datasets and is very similar to that described by Taylor for *Irgd*.

Line 108 "Earlier studies" sentence requires a reference, Khaminets et al 2010.

Line 114. The correct map of the chromosome 11 cluster of IRG genes, shown in Fig 1A was published as Fig 1 by Bekpen et al 2005 and was the first publication of this structure. The chromosomal nucleotide distances have changed slightly.

Line 115 The *Irgb6* locus was shown to contain two nearly identical copies by Bekpen et al 2005, where they were named as *Irgb6* and *Irgb6**. The present authors are therefore introducing a new nomenclature over a published nomenclature. They must justify this.

Line 115. *Irgb6* and *Irgb6** in fact differ by 4 non-coding nucleotides in the >1200 nucleotide-long *Irgb6* coding region. This is not "99%". Authors should be precise

Line 116 not "matched", "identical"

Line 117/118 No data presented on the genomic KOs. Fig 1A has no relevant information except the clear Western blot showing *Irgb6* deficiency. What were the KOs due to, nonsense mutations, premature stops, gene deletion? Are the two exons of the tandem decoy protein *Irgb5-b4* still present and unaffected? This is routine data to accompany a genomic KO, and must be presented, whether as main or supplementary data.

Lines 121-125 Data in Fig 1C represent % Toxos relative to cells not treated with IFN γ (calculated from luciferase yield). Thus the number is reduced to approximately 30% on the *Irgb6* and *b6+b10* KOs, while the wild-type causes reduction to 3-5%. Is this difference because there is residual IRG effector activity? It would be interesting to include a control of the *Irgm1/Irgm3* double KO, where effector activity may be more completely lost. This is important because the sense of the paper is that *Irgb6* is the key to the whole system, while residual 30% survival of the parasite allows for 70% destruction (or "clearance" as the authors prefer) in the *Irgb6* KO. To what can this destruction be attributed? Perhaps the combined activity of the other IRG effectors? This figure would be better if it included the a bar representing the parasites found in the absence of IFN γ at 100%.

Line 128/9: references for vacuolation and disruption of vacuoles.

Line 141. The word "regulates" carries inappropriate meanings. Better is "whether *Irgb6* is required for normal loading etc etc"

Lines 144-148. Hard to understand. Fig 1B shows there is apparently normal expression of the other IRG and GBP proteins. Better would be:

"the *Irgb6*-deficient MEFs displayed a significantly reduced accumulation of *Irga6*, *Gbp1*, *Gbp2*, *Gbp1-5* and *Irgb10* on the *T. gondii* PVM (Fig 2A-E). Fig 1B shows that this cannot be due to reduced protein expression."

Line 161 "*Irgb6* is capable of significantly mediating the bind (sic) of ubiquitin". This is a misleading form of words: again the correct term is "required for" the normal levels of binding of ubiquitin.

Line 170. Again, "is required for", not "universally regulates"

Line 173/4 and 182/183 The dependence of *Irgb6* localization behaviour on the presence of GMS regulator proteins was extensively studied by Hunn et al, 2008 in a different system.

Lines 198/199. This result was also reported by Hunn 2008.

Lines 224-230, Fig 4D: This is a very important claim, that only PI5P and PS were detected with the set of 5 detector proteins. Elsewhere in the paper, such labelling data does not rely, as here, on a single microscope image for each protein. It is always possible to find vacuoles negative for any marker you look for. This result must be quantitated blind by an innocent observer, and the quantitations presented within Fig 4D

Line 231-245 The data presented here is very much open to question because, as is apparent in the Materials and Methods (Line 541) though not explicitly stated in this paragraph, the experimental incubations did not include nucleotides, thus the apparent activity of *Irgb6* in these assays is due to the apoprotein alone. This important point is revealed only in the Discussion (Line 368 et seq), where the discrepancy with the behaviour of GTPase-deficient proteins is mentioned. It

is therefore unclear what property of the protein is being studied in this assay.

Lines 249-251. The structure of Irga6 revealed several C-terminal helices, not just F and K. The longest and most C-terminal of all is in fact Helix L. However Helix K was already implicated in the specificity of intracellular organelle targeting of the three GMS proteins, Irgm1, m2 and m3 in Martens et al, 2004 and Martens and Howard 2006, as well as for palmitoylation by Henry and Taylor (2014) so this helix was an obvious target for the present authors. The choice of Helix F is not clear and should be explained.

Line 250. The crystal structure of Irga6 provides the general structure for mouse IRG proteins, including the definitions of helices F and K. The correct reference is therefore Ghosh et al 2004, not Bekpen or Man.

Lines 268-270. Yes, the binding appears reduced, but the specificity is still apparent. How is this explained, and how is the exposure of the filters normalized?

Discussion

Line 303: Discussion. Oddly, the increased susceptibility of Irgb6-deficient mice to *T. gondii* infection is not mentioned at all in the Discussion. Generally quite limited.

1) As you will see, the reviewers think that your work is of value to the field, pending satisfactory major revision. However, quite an extensive revision is needed. Reviewer #2 provides constructive input on how to strengthen your work by adding appropriate controls (KO generation; comparison to Irgm1/Irgm3 double deficient condition), changing the data representation and by considering the value of the tubulation and lipid vesicle disruption analyses in light of the fact that the effects are independent of nucleotide.

Response: We would like to thank the reviewers for their invaluable comments. We addressed all the comments in detail in this letter and in revised manuscript with new data. We have carried out additional experiments, as suggested by reviewer #2, which is *T. gondii* killing assay using Irgm1/m3 double-deficient MEFs. The novel data is included in the revised Figure 1D.

We decided to not include the data of the tubulation and lipid vesicle disruption analyses in the revised version, as also suggested by reviewer #2. These data (former Figure 5A-C, Figure 6H and Supplementary Figure 3C and D) are fully removed in the revised version.

2) Both reviewers also point out that major efforts are needed for re-writing the manuscript and reviewer #2 points you to the most critical parts. We would thus like to invite you to submit a revised version of your work, addressing the criticisms raised.

Response: We thank for the comments and concerns raised by both reviewers. In the revised manuscript, correction of grammatical errors and English improvement were carried done by a native English-speaker. We hope it now suitable for the publication in Life Science Alliance.

Response to Reviewer #1 (Comments to the Authors (Required)):

We would like to thank reviewer #1 for your precious time and invaluable comments. We have carefully addressed reviewer #1's concerns in the revised version to improve the manuscript.

The manuscript by Lee et al addresses the role of Irgb6 in resistance to Toxoplasma gondii. Prior work has shown that Irgb6 localizes to the T. gondii vacuole (PV) in cells, and that it does so early in the cascade of IRG proteins that assemble on the vacuole. The current work advances our concept of Irgb6 function in several important respects. The reviewers show that deletion of Irgb6 greatly impairs immune attack on the T. gondii PV in contrast to lesser roles for other GKS proteins, indicating that Irgb6 is not only a sentinel but a pivotal and essential factor in the IRG/GBP cell autonomous T. gondii resistance program. The investigators show that Irgb6 is able itself to drive tabulation/vesiculation of membrane vesicles. Further, Irgb6 seems to target PI5P-rich membranes, and it does so through basic residues in its amino terminus. Together, this assembles many aspects of the mechanism through which Irgb6 functions.

This work will make an important addition to the literature. The series of studies are logically designed, the manuscript well-constructed, and statistics applied appropriately. The main deficiency in the manuscript is that many sentences are poorly constructed grammatically, in some cases altering their meaning. Careful editing should be performed to improve the grammar and sentence structure.

Response: We thank for the comments and concerns raised by reviewer #1. In the revised manuscript, correction of grammatical errors and English improvement were carried done by a native English-speaking proofreader. We hope it now suitable for the publication.

Response to Reviewer #2 (Comments to the Authors (Required)):

We would like to thank reviewer #2 for your precious time and the detailed and invaluable comments. These comments are very instructive, and would be helpful to improve our manuscript. We have carefully addressed all the comments in this letter and in the revised version.

Brief Summary: The central advance in this paper is a CRISPR-Cas9 KO of the duplicated Irgb6 gene on Chromosome 11 of the mouse and the discovery and analysis of a strong disease susceptibility phenotype. Irgb6 is a member of the multigene Immunity-Related GTPase (IRG) protein family in the mouse that is required for resistance to a small group of intracellular pathogens including Toxoplasma gondii. They are subdivided into 3 functional classes, effectors, regulators and decoys, each with distinctive roles in the overall resistance mechanism, which is associated with breakdown of the parasitophorous vacuole membrane (PVM). Irgb6 is an effector. Until now, 2 effector IRG genes have been knocked out, Irga6 and Irgd, both with significant but small effects on resistance against Toxoplasma gondii, and neither KO has a large effect on the loading of the other effector IRG proteins onto the PVM. The KO of Irgb6 proves to have a large effect on resistance, the other effector IRG proteins reach the PVM inefficiently and the PVM remains intact. It was reported several years ago that the loading of IRG proteins onto the PVM is a structured process, with Irgb6 as the first loaded "pioneer". The present results indicate that indeed Irgb6 is the pioneer, and that this status is not just first in a time series but also required for the normal loading of the other effectors.

The lack of injury to the PVM in the Irgb6 KO unsurprisingly leads to failure of the downstream effects of the IRG protein mediated resistance mechanism, namely the arrival at the PV of the 65 kDa GBPs, ubiquitin and the p62 autophagy adaptor.

More novel are the experiments that appear to identify specificity for PI5P on the PVM as the target for Irgb6. This is the first time that the basis of the highly specific targeting of the largely cytosolic effector IRG proteins to the PVM has been analysed. However, in view of the fact that other membranes also express PI5P, the specificity of PVM targeting is not fully accounted for in this study and there is certainly room for more work here.

Response: We fully agree with the reviewer #2 that the mechanism by which Irgb6 can specifically recognize and bind PI5P on the PVM, but not on other membranes, has not been accurately understood in this work. Endogenous mechanisms such as regulatory IRG proteins may be associated with the specificity of PVM targeting, but this needs to be clarified further.

Addition of soluble Irgb6 to giant unilamellar vesicles (GUVs) leads to distortion and vesicular breakdown, perhaps suggesting something of the mode of action in causing vacuolar breakdown.

Response: We fully agree with the reviewer #2 that the role of Irgb6 in tubulation and disruption of lipid vesicles in the absence of nucleotides is not consistent with the immunofluorescence data showing an essential role of GTPase activity of Irgb6 in their loading on the PVM. However, one possibility is that GTPase activity of Irgb6 is not required for phospholipid binding and tubulation in the extracellular environment. It needs to be clarified further.

As suggested by the reviewer #2, we decided to exclude these data (former Figure 5A-C, Figure 6H and Supplementary Figure 3C and D) in the revised version.

Review:

General Comments.

The results in this paper are of considerable interest in the field and should be published. However, the study is absolutely central to my own interests and work, and I am therefore at risk of being overcritical, without, I hope, any conflicts of interest. I should also say that the work was reported at a recent meeting where I discussed it with the senior author. The editors should bear this in mind when considering my review.

The paper is not very well written, significantly unclear in places and the English is often faulty. It is important that it be read and edited by a "native English speaker" but it is always the clear and correct meaning of the science that must be the basis for language edits.

Response: We thank for the comments and concerns raised by reviewer #2. The revised manuscript has been edited by a native English speaker, and the language errors in our manuscript were corrected. We hope it now suitable for the publication.

I have made a large number of detailed points below. Here I would like to highlight a few of these that I consider of greatest importance.

Response: We many thank reviewer #2 for the detailed and invaluable comments. We have carefully addressed all the comments in this letter and in the revised version.

Introduction:

Nomenclature. The authors make correct use of the gene and protein names defined by Bekpen et al, but they continue to use the GKS and GMS nomenclatures which divide IRG proteins into two functional classes. We introduced this dichotomous nomenclature when we first characterized the IRG protein family (Bekpen et al, 2005), using it only as a shorthand for the different sequence families.

*Subsequently, we showed that the GMS subgroup have a unique and essential function as negative regulators for GKS proteins that bind to the PV and act as effectors. However, in this dichotomous system another structurally distinct group of IRG proteins, the "tandems" was ignored, or rather, included in the GKS group through having classical nucleotide binding sites. We have since shown that at least one of these tandem IRG proteins has yet a third function, neither effector nor regulator, but acts to distract kinases secreted by virulent strains of *T. gondii*, preventing them from phosphorylating and inactivating the effector IRG proteins. We now therefore call the tandem IRG genes "decoys" (Mueller and Howard 2016).*

Response: We included a "decoys" in the Introduction (Line 67) and the reference in the revised manuscript.

I therefore suggest that the authors stop using the rather clumsy GMS/GKS nomenclature, that carries no functional information, and prefer to use "effector IRG proteins" or "effectors" and "regulator IRG proteins" or "regulators" (and "decoys" but they don't have any of those in this paper).

Response: Following the reviewer #2 suggestion, we replaced “GKS-Irg” or “GMS-Irg” with “effector IRG proteins” or “regulator IRG proteins”, respectively, in the revised

manuscript.

Referencing. I have made a substantial number of proposals for referencing that I believe are more correct and relevant. I specifically urge that the authors check directly with the references themselves whether they can agree with these changes or additions.

Response: We fully agree with the reviewer #2 that more correct and relevant references should be cited for accuracy. As suggested by the reviewer #2, we have made the changes or additions of the references in the revised manuscript.

Active or passive targeting of IRG proteins to the PV. The authors confuse the process by which the IRG proteins reach the PV and their tendency to stay there once they have arrived. Only the latter is relevant to their observations on PI5P interactions. This must be sorted out.

Response: We agree with the reviewer #2 that our study shows Irgb6 tendency to stay on the PVM via PI5P interactions once they have arrived. To make this clearer, we made changes in the Introduction and Discussion in the revised manuscript.

Results:

Description and analysis of the Crispr.cas mutations. In view of the multiple uncertainties associated with the CRISPR-Cas9 method, the authors must present specific sequence data on the nature of mutations they have introduced and at least state whether the Irgb5-4 tandem (decoy) gene that lies between the two mutated Irgb6 genes, is still present and unmutated. Can of course be in supplementary.

Response: Following the reviewer #2 suggestion, we conducted sequencing analysis of both Irgb6* and Irgb6 genomic DNA from Irgb6-deficient MEFs. The data revealed an 891-bp deletion in coding region (position 172-1062) in both Irgb6 genes. This novel data is shown in the revised Supplementary Figure 1A and B.

As also suggested by reviewer #2, we conducted sequencing analysis of Irgb5-b4 tandem (decoy) gene after amplification of the coding regions from cDNA of IFN- γ -treated Irgb6-deficient MEFs by PCR. The data showed that the complete coding region

of *Irgb5-b4* gene (1-2535 bp) is normally present between the two mutated *Irgb6* genes. This novel data is shown in the revised Supplementary Figure 2A-D.

Loss of cell-autonomous resistance in Irgb6-deficient cells, Fig 1C. This figure shows the extent to which the resistance of IFN γ -induced cells is damaged in the Irgb6 KO cells. The presentation is misleading since the absence of resistance in cells not treated with IFN γ is only implied on the ordinate as "T. gondii numbers %". If untreated cells are 100%, then what Fig1 shows is that fully 70% of the IFN γ -dependent resistance is still present in the b6-deficient cells. This would be obvious if another column equal to 100% were introduced into the figure. As it stands, the impression given is that resistance is completely lost, but this is clearly not so. This is already confusing. However perhaps more important is that the authors themselves do not draw attention to the substantial residual resistance and do not comment on it.

Response: We have now modified the graph by adding another column equal to 100% (Con, untreated cells with IFN- γ). The data is now shown in the revised Figure 1C.

Is it due to action by other effector IRG proteins like Irga6, whose access to the PV is not completely inhibited in the Irgb6-deficient cells (Fig 2A,B). Or else what? In order to better assess the scale of the deficiency it would be interesting to compare the Irgb6 deficient cells with Irgm1/Irgm3 double-deficient cells, which lack two of the three regulators and are generally considered to be completely deficient in IRG-mediated resistance. I consider this an important issue.

Response: Following the reviewer #2 suggestion, we compared *T. gondii* killing activity between *Irgb6* deficiency and *Irgm1/m3* double-deficiency in MEFs. The data showed that about 73% of the IFN- γ -dependent resistance is still present in the *Irgb6*-deficient cells, whereas about 12% of the IFN- γ -dependent resistance is present in *Irgm1/m3* double-deficient cells. This novel data is shown in the revised Figure 1D.

As the reviewer #2 mentioned, this difference may be caused by residual other effector IRG proteins as well as GBP proteins in the *Irgb6*-deficient cells (in the revised Figure 2A-E).

PIP binding domain results Fig 4D. The single microscope images shown here are not adequate. These results must be quantitated for all 5 proteins.

Response: As suggested by reviewer #2, we quantified all 5 proteins from confocal microscopy analysis of three independent experiments. The quantification data is included in revised Figure 4E.

Tubulation and disruption of lipid vesicles, Fig 5A-C. Very surprisingly, these effects of co-incubating lipid vesicles with the purified Irgb6 protein are independent of nucleotide. Since this is discordant with the cellular data that clearly show nucleotide-dependence, it is unclear what the observed effects are due to. In principle, this is an interesting experiment, but the lack of nucleotide dependence means that its relevance to the function of Irgb6 is not clear. I recommend eliminating this result from the paper, unless it can be supported by evidence for nucleotide-dependence. I note that no data is presented for characteristics such as the purity of the "purified" Irgb6 protein or its functional integrity, as would normally be provided in support of such a biophysical experiment.

Response: As suggested by the reviewer #2, we decided to exclude these data (former Figure 5A-C, Figure 6H and Supplementary Figure 3C and D) in the revised version. In order to test the purity of the purified proteins (WT-Irgb6 and K275/R371-Irgb6), we conducted SDS-PAGE and Coomassie blue staining. The data displayed that both proteins were detected as a single bright band in each well of the gel corresponding to the expected product sizes. This novel data is shown in the revised Figure 5H.

Detailed comments:

Abstract:

The abstract is full of problems. I here suggest an improved text, but cannot commit to doing this much work for the whole paper

Response: We appreciate the reviewer #2 comment. As suggested by reviewer #2, we have now modified the former abstract with the below suggested text in the revised manuscript.

Suggested corrected text:

Toxoplasma gondii (T. gondii) is an obligate intracellular protozoan parasite capable of infecting all warm-blooded animals by ingestion. The organism enters host cells and resides in the cytoplasm in a membrane-bounded parasitophorous vacuole (PV). The induction of an interferon response enables IFN γ -inducible immunity-related GTPases (IRG proteins) to accumulate on the PV and restrict parasite growth. However, little is known about the mechanisms by which IRG proteins recognize and destroy the T. gondii PV. We here characterized the role of Irgb6, an IRG protein, in the cell-autonomous response against T. gondii, entailing the ubiquitination and breakdown of the vacuole. We show that Irgb6 is capable of binding a specific phospholipid present on the PV membrane, and that absence of Irgb6 results in reduced or absent targeting of other effector IRG proteins to the PV implying that Irgb6 has a role as pioneer in the complex process by which multiple IRG proteins access the PV. Irgb6 deficient mice are highly susceptible to infection with a strain of T. gondii avirulent in wild-type mice.

Notes:

The first two sentences unbundle the clumsy first sentence of the authors' abstract.

Parasites are NOT cleared by the IRG protein response; some parasites always survive and these are responsible for persistent infection in brain cysts. However the IRG protein attack undoubtedly restricts parasite growth.

Death of mice from a failure of the IRG system was shown many years ago to result from unrestrained growth of a normally avirulent T. gondii strain and high levels of inflammatory cytokines. I do not see the point of restating this here as it is not a new finding.

I find the last sentence of the authors' abstract is redundant.

Introduction

As everywhere, difficult text.

Line 55: I am unaware of any virus infection resisted by IRG proteins; references to the very important Chlamydia studies are missed completely.

Response: “Virus” has been deleted in the Introduction (Line 63 after revision), and we have cited Chlamydia studies (Al-Zeer et al., 2009; Coers et al., 2008) (Lines 63-64 after revision).

Line 57: "IRGs consist of three regulatory etc etc" No; the IRG protein family in mice consists of.....

Response: Yes, we did this change (Line 65 after revision).

Line 57: There are many more genes than described proteins. The documented expressed IRG effector proteins are only 4: Irga6, Irgb6, Irgb10 and Irgd. Better to write "There are 4 known expressed effector IRG proteins, Irga6, Irgb6, Irgb10 and Irgd"

Response: Yes, we did this change (Lines 67-68 after revision).

Line 61: GTPase activity has been demonstrated and published in detail only for Irga6 and this should be referenced (Uthaiiah et al 2003, Hunn et al 2008). An early important, but not yet confirmed, report from Taylor and colleagues (JBC 271, 20399-20405) reported GTPase activity from Irgm3 (IGTP), a "GMS" protein.

Response: We have cited the references (Hunn et al., 2008; Taylor et al., 1996; Uthaiiah et al., 2003) (Line 75 after revision).

Lines 63 - 68 are hard to understand. The three GMS proteins are regulators that maintain the effectors ("GKS") proteins in an inactive GDP-bound state, probably to prevent the latter from activating inappropriately on host cell membrane-bounded vesicular systems. In their absence, effector proteins form what are probably GTP-bound aggregates and are unable to interact with the T. gondii PV. Correct references please.

Response: We have replaced the part with suggested sentences (Lines 75-79 after revision). We apologize for the errors in the references that we now corrected (Coers, 2013; Haldar et al., 2013; Hunn and Howard, 2010; Hunn et al., 2008; Martens et al., 2004) (Lines 79-81 after revision).

Line 68,69: Is it really true that all 11 mouse GBPs have demonstrated GTPase activity? Having an apparently intact nucleotide-binding site is not the same as having GTPase activity.

Response: We have made the changes to “There are 11 members of the mouse GBP family, all of which have the conserved GTP binding motifs (Kresse et al., 2008).” (Lines 81-82 after revision).

Line 72. Add "vesiculated" with disrupted, to anticipate data in Fig 1D

Response: Yes, we did this change (Lines 85 after revision).

Line 72/73. Reference to Collazo et al is inappropriate here. Remove.

Response: We have removed the reference (Lines 86-87 after revision).

Lines 77-90 This paragraph confuses the mechanism by which IRG and GBP proteins initially encounter a PV, which at least in the case of IRG proteins, is almost certainly diffusion, and the mechanism that causes the IRG proteins to be retained at the vacuole, which is what is referred to in this paper.

Response: We agree with the reviewer #2 and have made the changes to “The mechanism by which IRG proteins access the *T. gondii* PV from the cytosolic compartments can be passive. This process depends on diffusion from the cytoplasmic pools rather than active transport involving TLR-mediated signaling pathways or microtubule networks (Khaminets et al., 2010). Although IRG proteins are localized on the PVM within a few minutes of *T. gondii* infection (Hunn et al., 2008; Khaminets et al., 2010), little is known about the mechanism by which IRG proteins recognize and destroy the PVM thus far. This process is important for IFN- γ -induced cell-autonomous immunity.” (Lines 90-97 after revision).

*Lines 90-101 appear to suggest that *Irgb6* is both necessary and sufficient for resistance against *T. gondii*. The experiments described in the paper show only that *Irgb6* is required for the full resistance response. Since both *Irga6* and *Irgd* have both also been shown to be*

required for wild-type levels of resistance it is clear that Irgb6 may well not be sufficient. There is no evidence that a mouse possessing only Irgb6 from the effector set could fully resist T. gondii. It would now be possible to prepare such a mouse and test the issue.

Response: We agree with the reviewer #2 and have made the changes to “Here, we aimed to determine the role of Irgb6 in the cell-autonomous response against *T. gondii*, mediating ubiquitination and disruption of the *T. gondii* PVM.” (Lines 99-100 after revision).

Results

Line 105 and elsewhere. Authors always use the word "clearance" to describe the cell-autonomous process of vacuole destruction, but the term is normally used to describe an animal free from infection, which is not the case for T. gondii. IFNg-induced resistance of cells to T. gondii is accompanied by necrotic death of the cells. Clearance is misleading and must be changed.

Response: We agree with the reviewer #2 that “clearance” is misleading and must be changed. We have therefore replaced “clearance” with “killing” or “killing activity” wherever applicable in the revised manuscript.

Line 106/107. It is wrong to refer to the susceptibility of Irga6 KO mice as "controversial". The reference Liesenfeld et al 2011 describes the data obtained by two independent laboratories using two independent KOs of Irga6. The survival data is essentially indistinguishable between the two datasets and is very similar to that described by Taylor for Irgd.

Response: Reviewer #2 is right in pointing out that Irga6 protects mice against *T. gondii* infection *in vivo*. We have made the changes to “Several studies using gene-deficient mice (Liesenfeld et al., 2011; Taylor et al., 2000; Taylor et al., 2007) have shown that Irgm1 (also called LRG-47), Irgm3 (IGTP), Irga6 (IIGP, IIGP1), and Irgd (IRG-47) have critical roles in the anti-*T. gondii* response.” (Lines 104-106 after revision).

Line 108 "Earlier studies" sentence requires a reference, Khaminets et al 2010.

Response: Yes, we did this change (Line 107 after revision).

Line 114. The correct map of the chromosome 11 cluster of IRG genes, shown in Fig 1A was published as Fig 1 by Bekpen et al 2005 and was the first publication of this structure. The chromosomal nucleotide distances have changed slightly.

Response: This difference might be because we used NCBI database in old Figure 1A. Based upon ENSEMBL database, we have now corrected the chromosomal nucleotide distances in revised Figure 1A.

Line 115 The Irgb6 locus was shown to contain two nearly identical copies by Bekpen et al 2005, where they were named as Irgb6 and Irgb6. The present authors are therefore introducing a new nomenclature over a published nomenclature. They must justify this.*

Response: We agree with the reviewer #2. We have therefore replaced “Irgb6-1” or “Irgb6-2” with “Irgb6*” or “Irgb6”, respectively, in revised version.

Line 115. Irgb6 and Irgb6 in fact differ by 4 non-coding nucleotides in the >1200 nucleotide-long Irgb6 coding region. This is not "99%". Authors should be precise*

Response: We have made the changes to “The Irgb6 locus in C57BL/6 mice contains two Irgb6 genes (Irgb6* and Irgb6), both of which encode identical amino acid sequences despite their nucleotide coding sequences differing at 4 positions (Bekpen et al., 2005).” (Lines 111-113 after revision).

Line 116 not "matched", "identical"

Response: Yes, we did this change (Line 112 after revision).

Line 117/118 No data presented on the genomic KOs. Fig 1A has no relevant information except the clear Western blot showing Irgb6 deficiency. What were the KOs due to, nonsense mutations, premature stops, gene deletion? Are the two exons of the tandem decoy protein Irgb5-b4 still present and unaffected? This is routine data to accompany a genomic KO, and

must be presented, whether as main or supplementary data.

Response: Following the reviewer #2 suggestion, we conducted sequencing analysis of both *Irgb6*^{*} and *Irgb6* genomic DNA from *Irgb6*-deficient MEFs. The data revealed an 891-bp deletion in coding region (position 172-1062) in both *Irgb6* genes. This novel data is shown in the revised Supplementary Figure 1A and B.

As also suggested by reviewer #2, we conducted sequencing analysis of *Irgb5-b4* tandem (decoy) gene after amplification of the coding regions from cDNA of IFN- γ -treated *Irgb6*-deficient MEFs by PCR. The data showed that the complete coding region of *Irgb5-b4* gene (1-2535 bp) is normally present between the two mutated *Irgb6* genes. This novel data is shown in the revised Supplementary Figure 2A-D.

*Lines 121-125 Data in Fig 1C represent % Toxos relative to cells not treated with IFN γ (calculated from luciferase yield). Thus the number is reduced to approximately 30% on the *Irgb6* and *b6+b10* KOs, while the wild-type causes reduction to 3-5%. Is this difference because there is residual IRG effector activity? It would be interesting to include a control of the *Irgm1/Irgm3* double KO, where effector activity may be more completely lost. This is important because the sense of the paper is that *Irgb6* is the key to the whole system, while residual 30% survival of the parasite allows for 70% destruction (or "clearance" as the authors prefer) in the *Irgb6* KO. To what can this destruction be attributed? Perhaps the combined activity of the other IRG effectors? This figure would be better if it included the a bar representing the parasites found in the absence of IFN γ at 100%.*

Response: Following the reviewer #2 suggestion, we compared *T. gondii* killing activity between *Irgb6* deficiency and *Irgm1/m3* double-deficiency in MEFs. The data showed that about 73% of the IFN- γ -dependent resistance is still present in the *Irgb6*-deficient cells, whereas about 12% of the IFN- γ -dependent resistance is present in *Irgm1/m3* double-deficient cells. This novel data is shown in the revised Figure 1D. As the reviewer #2 mentioned, this difference may be caused by residual other effector IRG proteins as well as GBP proteins in *Irgb6*-deficient cells (in the revised Figure 2A-E).

Line 128/9: references for vacuolation and disruption of vacuoles.

Response: We have included the reference in the revised manuscript (Lines 142-145).

Line 141. The word "regulates" carries inappropriate meanings. Better is "whether Irgb6 is required for normal loading etc etc"

Response: Yes, we did this change (Line 156 after revision).

*Lines 144-148. Hard to understand. Fig 1B shows there is apparently normal expression of the other IRG and GBP proteins. Better would be:
"the Irgb6-deficient MEFs displayed a significantly reduced accumulation of Irga6, Gbp1, Gbp2, Gbp1-5 and Irgb10 on the T. gondii PVM (Fig 2A-E). Fig 1B shows that this cannot be due to reduced protein expression."*

Response: We apologize for the errors. We have made the changes accordingly to reviewer #2's comment (Lines 158-161 after revision).

Line 161 "Irgb6 is capable of significantly mediating the bind (sic) of ubiquitin". This is a misleading form of words: again the correct term is "required for" the normal levels of binding of ubiquitin.

Response: Yes, we did this change (Line 176 after revision).

Line 170. Again, " is required for", not " universally regulates"

Response: Yes, we did this change (Line 185-187 after revision).

Line 173/4 and 182/183 The dependence of Irgb6 localization behaviour on the presence of GMS regulator proteins was extensively studied by Hunn et al, 2008 in a different system.

Response: We have made the changes to "Thus, we examined the contribution of regulator IRG proteins to controlling Irgb6 localization on the T. gondii PVM using Irgm1/m3 double-deficient MEFs. A previous study has reported that regulator IRG proteins are required for the localization of Irgb6 on the PVM in a different system (Hunn et al., 2008)." (Lines 199-202).

Lines 198/199. This result was also reported by Hunn 2008.

Response: We have included the reference in the revised manuscript (Lines 215-217). We have made the changes to “However, the K69A-Irgb6 mutant in WT cells was found to localize to the *T. gondii* PVM, as described previously (Hunn et al., 2008).”

Lines 224-230, Fig 4D: This is a very important claim, that only PI5P and PS were detected with the set of 5 detector proteins. Elsewhere in the paper, such labelling data does not rely, as here, on a single microscope image for each protein. It is always possible to find vacuoles negative for any marker you look for. This result must be quantitated blind by an innocent observer, and the quantitations presented within Fig 4D

Response: As suggested by reviewer #2, we quantified all 5 proteins from confocal microscopy analysis of three independent experiments. The quantification data is included in revised Figure 4E (Line 247-252).

Line 231-245 The data presented here is very much open to question because, as is apparent in the Materials and Methods (Line 541) though not explicitly stated in this paragraph, the experimental incubations did not include nucleotides, thus the apparent activity of Irgb6 in these assays is due to the apoprotein alone. This important point is revealed only in the Discussion (Line 368 et seq), where the discrepancy with the behaviour of GTPase-deficient proteins is mentioned. It is therefore unclear what property of the protein is being studied in this assay.

Response: As suggested by the reviewer #2, we decided to exclude these data (former Figure 5A-C, Figure 6H and Supplementary Figure 3C and D) in the revised version.

Lines 249-251. The structure of Irga6 revealed several C-terminal helices, not just F and K. The longest and most C-terminal of all is in fact Helix L. However Helix K was already implicated in the specificity of intracellular organelle targeting of the three GMS proteins, Irgm1, m2 and m3 in Martens et al, 2004 and Martens and Howard 2006, as well as for palmitoylation by Henry and Taylor (2014) so this helix was an obvious target for the present authors. The choice of Helix F is not clear and should be explained.

Response: We thank reviewer #2 for the suggestion to include these references. We have made the changes to “The C-terminal α -K helix of regulatory IRG proteins seems to be crucial for the specificity of intracellular organelle targeting (Henry et al., 2014; Martens and Howard, 2006; Martens et al., 2004) or for targeting to the mycobacterial phagosome (Tiwari et al., 2009). A recent study showed that the C-terminal α -helical domains (especially α F and α K) of Irgb10 were predicted to be required for Irgb10 antimicrobial action involving intracellular bacterial membrane targeting (Man et al., 2016). Irgb6 is predicted to possess two C-terminal α -helical domains (α F and α K) based on the crystal structure analysis of Irga6, which provides the general structure for mouse IRG proteins (Fig 5A) (Ghosh et al., 2004). We therefore focused on two C-terminal α -helical domains (α F and α K) in Irgb6.” (Lines 257-266 after revision).

Line 250. The crystal structure of Irga6 provides the general structure for mouse IRG proteins, including the definitions of helices F and K. The correct reference is therefore Ghosh et al 2004, not Bekpen or Man.

Response: Yes, we have made the changes to “Irgb6 is predicted to possess two C-terminal α -helical domains (α F and α K) based on the crystal structure analysis of Irga6, which provides the general structure for mouse IRG proteins (Fig 5A) (Ghosh et al., 2004).” (Lines 263-265 after revision).

Lines 268-270. Yes, the binding appears reduced, but the specificity is still apparent. How is this explained, and how is the exposure of the filters normalized?

Response: We agree with reviewer #2 comments that PIP binding ability of K275/R371-Irgb6 was greatly reduced, but the specificity was still apparent. One possibility is that high protein concentrations (0.5 μ g/ml) used in this experiment can lead to a higher sensitivity to phospholipid bindings when considered with the physiological levels of Irgb6.

For a protein–lipid overlay assay using the purified proteins (WT-Irgb6 and K275/R371-Irgb6), the signals were detected simultaneously to ensure equal exposure times. The image of Fig. 5G has a line between WT-Irgb6 and K275/R371-Irgb6, however, the line was retrofit on the original image of a film showing that the two PIPstrip membranes were equally exposed and simultaneously developed (Right figure). We have now included the description in the RESULTS of the revised manuscript (Lines 286-287).

In order to test the purity of the purified proteins (WT-Irgb6 and K275/R371-Irgb6), we conducted SDS-PAGE and Coomassie blue staining. The data displayed that both proteins were detected as a single bright band in each well of the gel corresponding to the expected product sizes. This novel data is shown in the revised Figure 5H (Lines 287-290).

Discussion

Line 303: Discussion. Oddly, the increased susceptibility of Irgb6-deficient mice to T. gondii infection is not mentioned at all in the Discussion. Generally quite limited.

Response: We thank reviewer #2 for the accurate reading of the manuscript. We apologize for these errors. The in vivo experiments were included in the Discussion of the revised manuscript (Lines 393-402).

November 28, 2019

RE: Life Science Alliance Manuscript #LSA-2019-00549-TR

Prof. Masahiro Yamamoto
Graduate School of Medicine, Osaka University
Osaka University
2-2 Yamadaoka
Suita city, Osaka 565-0871
Japan

Dear Dr. Yamamoto,

Thank you for submitting your revised manuscript entitled "Initial phospholipid-dependent Irgb6 targeting to *Toxoplasma gondii* vacuoles mediates host defense". I now assessed your revised version and also spoke with the original reviewer #2 about it. We appreciate the introduced changes and would thus be happy to publish your paper in Life Science Alliance pending final revisions necessary to meet our requirements:

- I suggest to slightly further edit the abstract:

Toxoplasma gondii (*T. gondii*) is an obligate intracellular protozoan parasite capable of infecting warm-blooded animals by ingestion. The organism enters host cells and resides in the cytoplasm in a membrane-bound parasitophorous vacuole (PV). Inducing an interferon response enables IFN- γ -inducible immunity-related GTPase (IRG protein) to accumulate on the PV and to restrict parasite growth. However, little is known about the mechanisms by which IRG proteins recognize and destroy the *T. gondii* PV. We characterized the role of IRG protein Irgb6 in the cell-autonomous response against *T. gondii*, which involves vacuole ubiquitination and breakdown. We show that Irgb6 is capable of binding a specific phospholipid on the PV membrane. Furthermore, the absence of Irgb6 causes reduced targeting of other effector IRG proteins to the PV. This suggests that Irgb6 has a role as pioneer in the process by which multiple IRG proteins access the PV. Irgb6-deficient mice are highly susceptible to infection by a strain of *T. gondii* avirulent in wild-type mice.

- Please upload the supplementary figures as individual files. The supplementary table should get moved to the main manuscript file; the main manuscript file needs to get provided in word docx format

- The insets in Fig 1E and F are difficult to see, please change the color

- The inset in Fig 1E does not match exactly the outlined area, please fix

- Fig 4&5: clarification needed regarding experimental procedure and displayed representatives for the PIP strip analyses (WT condition); please provide this clarification in a written format (eg cover letter) to us

- Please mention the statistical test used next to the p-values in the figure legends

A. FINAL FILES:

B. MANUSCRIPT ORGANIZATION AND FORMATTING:

Sincerely,

We would like to thank editor for your invaluable comments. We addressed all the comments in detail in this letter and in revised manuscript.

- I suggest to slightly further edit the abstract:

Toxoplasma gondii (T. gondii) is an obligate intracellular protozoan parasite capable of infecting warm-blooded animals by ingestion. The organism enters host cells and resides in the cytoplasm in a membrane-bound parasitophorous vacuole (PV). Inducing an interferon response enables IFN-g-inducible immunity-related GTPase (IRG protein) to accumulate on the PV and to restrict parasite growth. However, little is known about the mechanisms by which IRG proteins recognize and destroy the T. gondii PV. We characterized the role of IRG protein Irgb6 in the cell-autonomous response against T. gondii, which involves vacuole ubiquitination and breakdown. We show that Irgb6 is capable of binding a specific phospholipid on the PV membrane. Furthermore, the absence of Irgb6 causes reduced targeting of other effector IRG proteins to the PV. This suggests that Irgb6 has a role as pioneer in the process by which multiple IRG proteins access the PV. Irgb6-deficient mice are highly susceptible to infection by a strain of T. gondii avirulent in wild-type mice.

Response: We appreciate the editor comment. As suggested by editor, we have now modified the former abstract with the above suggested text in the revised manuscript.

- Please upload the supplementary figures as individual files. The supplementary table should get moved to the main manuscript file; the main manuscript file needs to get provided in word docx format

Response: As suggested by editor, we uploaded the supplementary figures as individual files and provided the main manuscript file as word docx format. The supplementary table was now included in the revised manuscript file.

- The insets in Fig 1E and F are difficult to see, please change the color

Response: As suggested by the editor, we changed the color of insets in revised Figure 1E and F.

- The inset in Fig 1E does not match exactly the outlined area, please fix

Response: We apologize for the errors. We did this change in revised Figure 1E.

- Fig 4&5: clarification needed regarding experimental procedure and displayed representatives for the PIP strip analyses (WT condition); please provide this clarification in a written format (eg cover letter) to us

Response: As suggested by editor, we have now provided the clarification regarding experimental procedure for the PIP strip analyses (WT condition) in the below.

To investigate lipid-Irgb6 protein interactions, we conducted a protein-lipid overlay assay as following:

- 1) The PIP strip membrane (P-6001, Echelon Biosciences Inc) was blocked with TBS-T (0.1% Tween-20) + 3% BSA (fatty acid free) for 1 hr at 25 °C. Gently agitate.
- 2) Discard blocking buffer and add 0.5 µg/ml purified recombinant protein Irgb6-His tagged in 1.5 mL TBS-T + 3% BSA. Incubate for overnight at 4 °C.
- 3) Discard the protein solution and wash with TBS-T five times with gentle agitation for 5 min each.
- 4) Discard wash buffer and add 1.5 µL anti-His-HRP antibody in 1.5 mL TBS-T + 3% BSA. Incubate the membrane for 1 hr at 25 °C with gentle agitation.
- 5) Discard the antibody solution and wash with TBS-T six times with gentle agitation for 10 min each.
- 6) Discard wash buffer and detect the bound protein by ECL development followed by ImageQuant LAS 4000 (GE Healthcare).
- 7) The displayed images in the Figure 4 & 5 were used from ImageQuant LAS 4000.

The detailed procedure above is added in the Materials and Methods section in the revised manuscript.

- Please mention the statistical test used next to the p-values in the figure legends

Response: Following the editor suggestion, we have included the statistical test in the Figure legends (Lines 895, 915, 974, 985, 1016, and 1034).

December 2, 2019

RE: Life Science Alliance Manuscript #LSA-2019-00549-TRR

Prof. Masahiro Yamamoto
Research Institute for Microbial Diseases
Osaka University
3-1, Yamadaoka
Suita city, Osaka 565-0871
Japan

Dear Dr. Yamamoto,

Thank you for submitting your Research Article entitled "Initial phospholipid-dependent Irgb6 targeting to *Toxoplasma gondii* vacuoles mediates host defense". It is a pleasure to let you know that your manuscript is now accepted for publication in Life Science Alliance. Congratulations on this interesting work.

DISTRIBUTION OF MATERIALS:

Again, congratulations on a very nice paper. I hope you found the review process to be constructive and are pleased with how the manuscript was handled editorially. We look forward to future exciting submissions from your lab.

Sincerely,
